# Intrinsic Task Symmetry Drives Generalization in Algorithmic Tasks

**Hyeonbin Hwang** [1]   **Yeachan Park** [2]

## Abstract

Grokking, a sudden transition from memorization to generalization, has been closely linked to the emergence of low-dimensional representations; yet the mechanism driving this organization remains elusive. Here, we propose that intrinsic task symmetries are the key drivers of grokking, inducing structured geometries in representation space. Our analysis reveals a consistent three-stage training dynamic: (i) data memorization, (ii) intrinsic symmetry acquisition, and (iii) geometric organization. We show that generalization emerges during the symmetry acquisition phase, and subsequently the embedding space organizes into a low-dimensional structured geometry. We validate this mechanism across diverse algorithmic domains, spanning algebraic (modular arithmetic), structural (graph metric completion), and relational (comparison) reasoning tasks. Leveraging these insights, we formulate a symmetry-based criterion for generalization and propose symmetry- and geometry-prompting training strategies that can accelerate generalization. Together, our results establish intrinsic symmetry as a central mechanism enabling neural networks to move beyond memorization and achieve robust algorithmic reasoning.

## 1. Introduction

***What enables generalization in neural networks?*** This remains as the central open question in deep learning. State-of-the-art models show remarkable proficiency on tasks requiring intricate reasoning and abstraction, yet what they actually learn and under what conditions those solutions extend beyond the observed samples remains elusive. As a result, grokking (Power et al., 2022) has emerged as a particularly revealing testbed for studying generalization, characterized by a delayed yet abrupt transition from memorization to generalization.

Prevailing explanations for grokking such as implicit biases (Lyu et al., 2024), regularization norms (Liu et al., 2023), and circuit efficiency (Varma et al., 2023) converge on the observation that generalization arises when representations collapse into low-dimensional, structured manifolds (Nanda et al., 2023; Zheng et al., 2024). However, while *optimization*-centric explanations like weight decay account for the *tendency* toward compression, they alone cannot explain the *specific form* of the resulting geometry. Even task-specific circuits in tasks like modular addition (Nanda et al., 2023; Zhong et al., 2023) cannot reliably provide a generic explanation as to **why** the model chooses to solve the problem using such algorithm.

In this paper, we propose that **intrinsic task symmetry is the key driver of generalization**, determining the specific shape of the learned geometry in algorithmic tasks. To validate this hypothesis, we extend our analysis beyond the standard modular arithmetic testbed to a diverse suite of algebraic, structural, and relational domains.

By tracking the specific symmetries governing each of these tasks, we identify a consistent three-stage training dynamic: (i) Memorization, (ii) Symmetry Acquisition, and (iii) Geometric Organization. In doing so, our framework complements prior grokking phase characterizations (Nanda et al., 2023) by providing a causal explanation for *why* specific representations emerge from a data-centric view.

We demonstrate that generalization emerges alongside symmetry acquisition, identifying a *soft threshold* of symmetry violation below which generalization reliably occurs. We further show that intrinsic symmetry, together with low-rank representations, gives rise to a specific geometric organization of the embedding space. Building on these observations, we validate intrinsic symmetry as a key driver of generalization by demonstrating that symmetry- and geometry-prompting strategies can accelerate the grokking process. To summarize, our contributions are as follows:

- **Intrinsic Task Symmetry as a Key Driver of Generalization:** We establish intrinsic task symmetries as the primary driver of generalization in algorithmic tasks, and empirically investigate this mechanism across three domains: algebraic (modular arithmetic), structural (graph metric completion), and relational (comparison).

---

[1]KAIST [2]Sejong University. Correspondence to: Yeachan Park <ychpark@sejong.ac.kr>.

*Proceedings of the 43rd International Conference on Machine Learning*, Seoul, South Korea. PMLR 306, 2026. Copyright 2026 by the author(s).

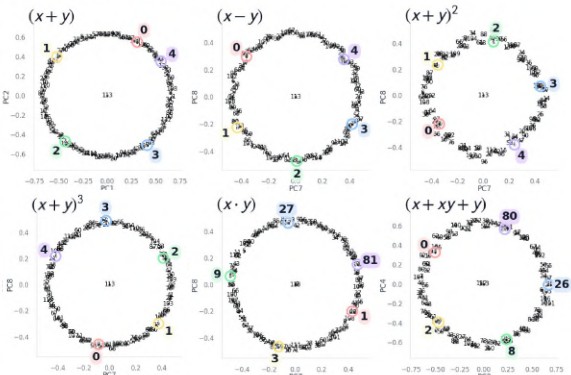

*Figure 1.* Modular Arithmetics embeddings

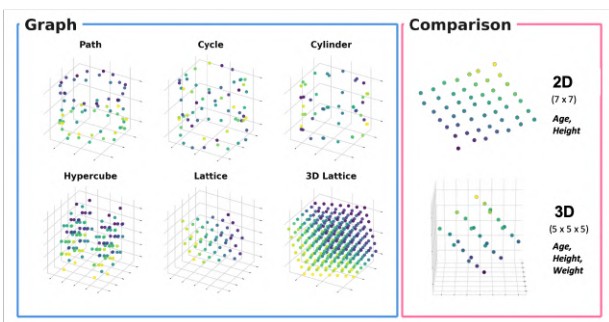

*Figure 2.* Graph Metric Completion and Comparison Embeddings.

- **Symmetry-aware Training Dynamics of *grokking*:**
  We propose a generic three-stage training dynamic for
  grokking: (i) *Memorization*, (ii) *Symmetry Acquisition*,
  and (iii) *Geometric Organization*. We show that generalization emerges during the *Symmetry Acquisition*
  phase, which can be tracked using a proposed criterion,
  and that intrinsic symmetry, together with low-rank representations, steers the geometric organization of the
  embedding space.

- **Accelerating Generalization via Symmetry Prompting:** Leveraging these insights, we show that symmetry-
  and geometry-prompting strategies can accelerate
  grokking, providing evidence for a causal role of symmetry alignment in generalization.

## 2. Related Works

**Grokking**, which was first observed in modular arithmetic
tasks (Power et al., 2022), has prompted extensive investigation into the mechanisms of delayed generalization. Prevailing explanations range from theoretical analyses of simple
models (Gromov, 2023; Žunkovič & Ilievski, 2024; Lyu
et al., 2024; Mohamadi et al., 2024; Gu et al., 2025; Levi
et al., 2024; Nam et al., 2025) to optimization-centric views
that cite small weight norms (Liu et al., 2023), low-rank
biases (Notsawo et al., 2025), initialization regimes (Kumar
et al., 2024), circuit competition dynamics (Merrill et al.;
Varma et al., 2023; Huang et al., 2024), or numerical instability (Prieto et al., 2025).

From a mechanistic perspective, many works have clarified
how models generalize, tying grokking to the emergence of
highly structured representations. For example, Liu et al.
(2022) and Zheng et al. (2024) argue that a sharp reorganization of representational geometry acts as the primary
signal of generalization, while Tian (2025) studies the staged
emergence of generalizable features during grokking. In
the specific case of modular arithmetic, this representations
manifests geometrically as circular or helical structures in

the embedding space, often interpreted through Fourier analysis (Nanda et al., 2023; Zhong et al., 2023). Notably, these
geometric mechanisms also appear in pre-trained language
models (Zhou et al., 2024; Kantamneni & Tegmark, 2025).

While the emergence of these geometric representations
is well documented, the question of *why* they take these
specific forms remains largely unanswered, often attributed
generically to weight decay (Nanda et al., 2023; Zhu et al.,
2024). A complementary data-centric perspective instead
emphasizes training-test distribution shift (Liu et al., 2023;
Wang et al., 2024; Gu et al., 2025). Notably, Chang et al.
(2025) formalize this through functional equivalence for
predicting coverage in compositional generalization, while
other theoretical studies link the learning of invariance to
generalization bounds (Xu et al., 2020; Benton et al., 2020).

Recent work has also begun to connect symmetry, symbolic structure, and representation geometry more directly.
Tian (2026); Wang & Wang (2025) develop a theoretical
framework in which gradient-based training can recover
algebraic symbolic structures under geometric constraints
such as group invariance. From a broader algorithmic-agent
perspective, Ruffini et al. (2025) argue that agents tracking
structured data can inherit the symmetries of the underlying
generative process, leading to reduced, manifold-like organization. Our work advances this discourse by proposing that,
across diverse algorithmic domains, intrinsic task symmetry
is the underlying invariance that causally drives generalization and shapes the resulting representation geometry.

Lastly, parallel to mechanistic inquiries, other works focus
on accelerating the onset of generalization. These include
transfer learning from pretrained or weaker models (Furuta
et al., 2024; Xu et al., 2025; Park et al., 2025), optimization
techniques like *GrokFast* that amplify slow-varying gradients (Lee et al., 2024), and the enforcement of inductive
biases such as commutativity in modular arithmetic (Tan &
Huang, 2023; Park et al., 2025).

## 3. General Setup

### 3.1. Modular Arithmetic

Following Power et al. (2022), we first evaluate generalization on algorithmic tasks over the ring of integers modulo $p$, denoted as $\mathcal{X} = \mathbb{Z}_p = \{0, 1, \ldots, p-1\}$. We consider a suite of six tasks ranging from simple linear groups to higher-order polynomials. Formally, given inputs $x, y \in \mathbb{Z}_p$, the network must predict the target $z$ defined by:

$$T_1 = (x + y) \quad T_2 = (x - y) \quad T_3 = (x + y)^2$$
$$T_4 = (x + y)^3 \quad T_5 = x \times y \quad T_6 = (x + xy + y)$$

### 3.2. Graph Metric Completion

While modular arithmetic has served as a popular testbed for grokking, these algebraic operations can be viewed geometrically as movements on a specific graph topology: the cycle graph $C_p$ of the ring $\mathbb{Z}_p$. In this light, we extend the setup from simple cycles to arbitrary graph structures with explicit geometries.

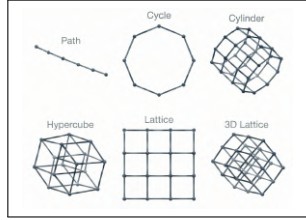

*Figure 3.* Type of Graphs.

We introduce the **Graph Metric Completion**[1] task, which aims to recover the global topological structure of the graph by inferring the full shortest-path distance matrix $D$. Specifically, given two vertices $u, v \in V$ in a graph $G = (V, E)$, the network is tasked with predicting the geodesic distance $d(u, v)$. In our experiments, we consider six distinct geometries: Path, Cycle, Cylinder, Hypercube, Lattice, 3D Lattice (Figure 3).

### 3.3. Comparison task

Now, we strip away the metric scaffolding provided in the graph task and investigate whether geometry still emerges under strictly *local* supervision. We adopt the **Comparison Task**, where the model must infer global structure solely from binary pairwise relations ($x \succ y$). To probe the limits of geometric representation, we focus on transitive regimes and evaluate performance in both 2D and 3D attribute spaces.

**Problem setup.** We instantiate the comparison task in a form analogous to a small knowledge graph, following prior works (Wang et al., 2024; Allen-Zhu & Li, 2024). Each entity represents a person characterized by latent scalar attributes, with dimensionality two or three depending on

---

[1]This task is the discrete analogue of the *Euclidean Distance Matrix Completion* problem.

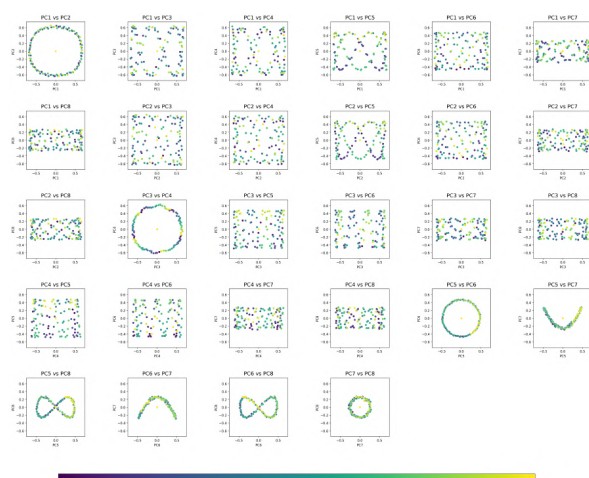

*Figure 4.* Full PCA visualization of modular addition embeddings.

the regime. Supervision is provided exclusively through relative order relations between pairs of entities, without access to absolute attribute values.

Let $\mathcal{X}$ denote the entity vocabulary and $\mathcal{R}$ the relation vocabulary:

$$\mathcal{X} = \{\, e_i = (a_i, h_i, w_i) \mid a_i, h_i, w_i \in \{1, 2, \ldots, m\} \,\},$$
$$\mathcal{R} = \{\, r^x_{\bowtie} \mid x \in \{\text{age}, \text{height}, \text{weight}\}, \bowtie \in \{<, =, >\} \,\}.$$

Then, each training example is a triple $(e_i, r, e_j)$ with label $y \in \{0, 1\}$, indicating whether the relation $r$ holds between $e_i$ and $e_j$. In natural language, this corresponds to sentences such as "John is older than Mary" or "Alice is taller than Bob".

## 4. Motivation : Structured geometry in embedding space

Across the tasks introduced above, our central goal is to understand how generalization relates to the geometric structure of learned embeddings. To this end, we analyze token embeddings using principal component analysis (PCA) and find that successful generalization consistently coincides with the emergence of structured, low-dimensional organization in embedding space.

In modular arithmetic tasks, the learned embeddings exhibit particularly clear geometric structure. As shown in Figure 1, 2D PCA projections reveal approximately circular patterns, consistent with prior observations in modular addition tasks (Nanda et al., 2023; Kantamneni & Tegmark, 2025). When extended to higher-dimensional PC subspaces, the embeddings organize into a helical geometry: different PC pairs exhibit circular projections with systematic phase shifts. An explanation of this structure is provided in Appendix I.

Similar geometric organization arises beyond modular arith-

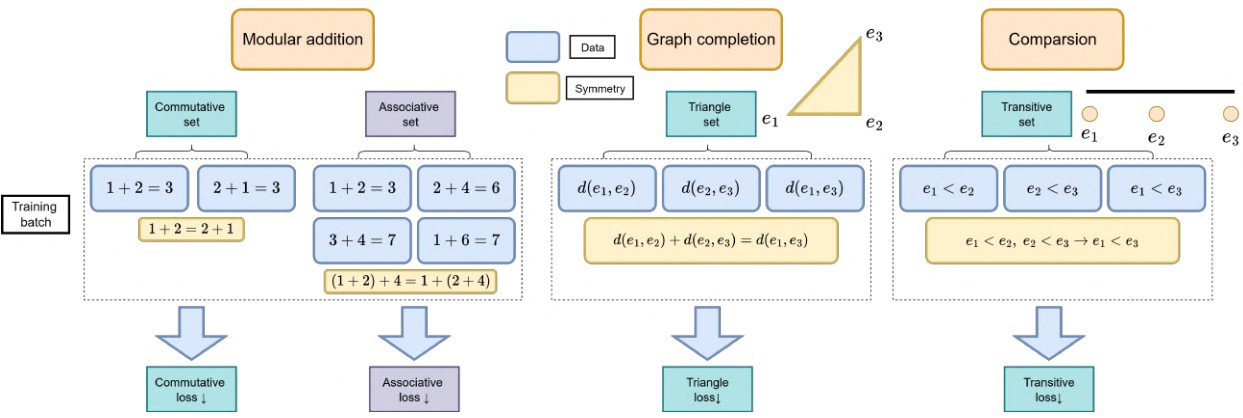

*Figure 5.* Symmetric patterns within a dataset lead the network to learn intrinsic symmetry.

metic. In graph completion and comparison tasks, the learned embeddings recover geometric structures that reflect the relational or ordering properties of the task, as illustrated in Figure 2. These observations suggest that task-aligned geometric structure is a general property of embeddings that support generalization in algorithmic tasks.

Related connections between embedding geometry and generalization have been noted in prior work (Liu et al., 2022). While such geometry helps explain how generalization is expressed in learned representations, the mechanism by which these structures emerge remains unclear. In Section 7, we argue that intrinsic task symmetries provide a principled driver for the emergence of structured embedding geometry. Additional PCA visualizations are provided in Appendix A.2.

## 5. Intrinsic Symmetries in algorithmic tasks

We next identify intrinsic symmetries present in algorithmic tasks, which are illustrated in Figure 5.

**Symmetry in modular arithmetic** In modular addition, there are two symmetries, commutativity and associativity:

$$\text{Commutativity: } x + y = y + x,$$

$$\text{Associativity: } x + (y + z) = (x + y) + z.$$

These two properties are basic properties in arithmetic operations. We also note that other operations have similar symmetries which we present in Table 1.

**Symmetry in graph metric completion task** In graph metric completion task, there exists the symmetry of the triangle equality. Let $e_1, e_2, e_3$ be vertices of the graph. If $e_2$ is placed on the shortest path between $e_1$ and $e_3$, then the following triangle equality holds:

$$\text{Triangular symmetry: } d(e_1, e_2) + d(e_2, e_3) = d(e_1, e_3).$$

*Table 1.* Symmetries in modular operations. Here $x^+ = x + 1$.

| Task | Commutativity proxy | Associativity proxy |
|---|---|---|
| $x + y$ | $x + y = y + x$ | $(x + y) + z = x + (y + z)$ |
| $x - y$ | $x - (-y) = y - (-x)$ | $\begin{aligned}(x - y) - z \\ = x - (y - (-z))\end{aligned}$ |
| $(x + y)^\alpha$ | $(x + y)^\alpha = (y + x)^\alpha$ | $\begin{aligned}((x + y) + z)^\alpha \\ = (x + (y + z))^\alpha\end{aligned}$ |
| $x \times y$ | $x \times y = y \times x$ | $\begin{aligned}(x \times y) \times z \\ = x \times (y \times z)\end{aligned}$ |
| $x + xy + y$ | $\begin{aligned}x^+ y^+ - 1 \\ = y^+ x^+ - 1\end{aligned}$ | $\begin{aligned}(x^+ y^+) z^+ - 1 \\ = x^+ (y^+ z^+) - 1\end{aligned}$ |

We clarify that the triangular symmetry holds only if $e_2$ is placed on the shortest path between $e_1$ and $e_3$.

**Symmetry in comparison task** In comparison task, there exists the transitivity symmetry. Let $e_1, e_2, e_3$ be token entities in the comparison tasks. Suppose we have $e_1 < e_2$ and $e_2 < e_3$ in the training batch. Then by transitivity, we also have $e_1 < e_3$ and we call it transitive symmetry.

$$\text{Transitivity : } e_1 < e_2, \quad e_2 < e_3 \quad \rightarrow \quad e_1 < e_3.$$

Details on how each metric can be computed are provided in Appendix A.

## 6. Symmetry-driven generalization

Having identified the intrinsic symmetries underlying each task, we now examine their functional role during training.

### 6.1. Three-Stage Training Dynamics

A common intuition is that once the network perfectly fits the training data, there is little left to learn and training effectively terminates. However, when intrinsic symmetries are present in the task, learning can continue even after

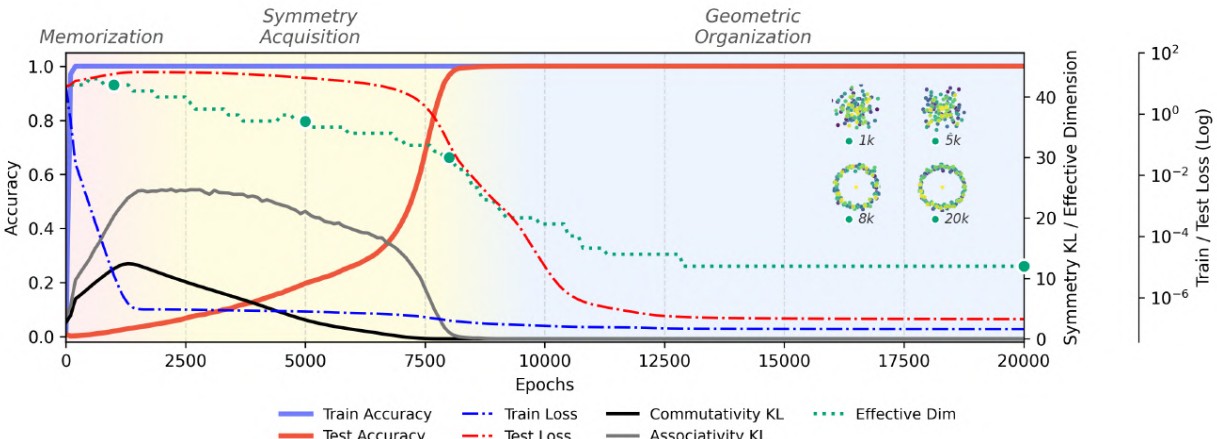

*Figure 6.* Three-stage dynamic of *grokking* for modular addition task. **(1) Memorization**: The model first *memorizes* as it quickly reaches perfect train accuracy. **(2) Symmetry Acquisition** Then with the symmetry criterion decreasing to certain threshold, perfect test accuracy is reached.**(3) Geometric Organization** Finally, as the representation collapses into lower-dimension, embedding forms a geometric organization.

memorizing the training samples by acquiring the intrinsic symmetry while maintaining low training loss. For example, in modular addition, if the training batch contains commutative pairs or associative tuples, the network is exposed to algebraic symmetries of addition. By repeatedly encountering symmetry patterns, the network can implicitly learn these symmetries beyond memorizing individual samples. Analogous mechanisms arise in other algorithmic tasks, where additional structural information embedded in the data enables the network to generalize. We summarize the intrinsic symmetries present in the considered algorithmic tasks in Figure 5.

Based on these observations, we propose that training proceeds through three distinct stages: (i) data memorization, (ii) intrinsic symmetry acquisition, and (iii) geometric organization. Figure 6 illustrates this three-stage training dynamics.

During the data memorization stage, the network fits the training samples by minimizing the training loss. Even after achieving the perfect training accuracy, this process may continue by reducing remaining training loss without yielding generalization.

Following sufficient memorization, the symmetry acquisition stage begins. In contrast to memorization stage, this stage is characterized by a qualitative change: the network begins to learn the intrinsic task symmetries, leading to improved performance on unseen data. When symmetry acquisition is sufficiently complete, generalization emerges.

Once symmetry acquisition stabilizes, the geometric organization stage begins. At this point, weight decay becomes the dominant influence on training dynamics. By penalizing weight norm, regularization promotes low-rank and

low-dimensional representations that are compatible with the learned symmetries, resulting in structured embedding geometries and improved stability. While regularization is active throughout training, its effect becomes most visible after symmetry constraints have reduced the representational dimensions.

Training dynamics for additional algorithmic tasks are provided in Appendix A. Although the precise timing and sharpness of stage transitions vary across tasks, we consistently observe that generalization coincides with symmetry acquisition rather than continued memorization, supporting the view that symmetry learning is a mechanism distinct from overfitting.

We further support this stage-wise picture through a gradient alignment analysis in Appendix B. During the symmetry acquisition stage, the optimizer direction becomes positively aligned with the gradients of the symmetry constraints, suggesting that training implicitly follows directions that reduce symmetry violations and promote generalization.

### 6.2. Symmetry-based criterion

Our observations indicate that generalization emerges when a network acquires an understanding of the intrinsic symmetries of the task. We therefore argue that symmetry alignment is a key factor underlying generalization. To make this claim testable, we introduce a symmetry-based criterion for generalization: once a model sufficiently aligns with the intrinsic task symmetry, it reliably generalizes beyond the training distribution.

As shown in Figure 7, we observe a soft threshold behavior in the symmetry metric, below which generalization consistently occurs across different runs. While the transition is

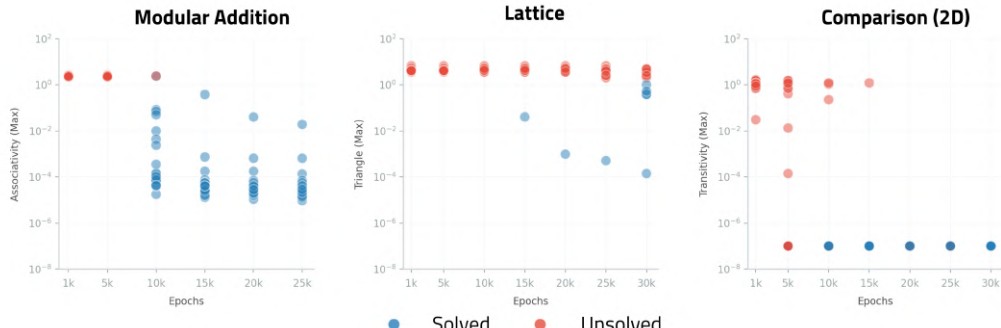

Figure 7. Symmetry Metric vs. Test Accuracy (20 seeds). With a soft threshold, the test accuracy achieves 1.0 as the symmetry metric minimizes.

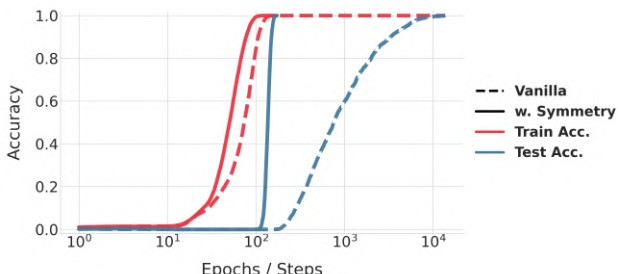

Figure 8. We train a two-layer MLP on modular addition using an 80% training fraction, *without* weight decay.

not perfectly sharp, models that reduce symmetry violation past this threshold reliably generalize, whereas models that remain above it do not. This behavior supports the use of symmetry alignment as a predictive criterion for generalization. Additional results for symmetry-based criteria on other algorithmic tasks are provided in Appendix C.

### 6.3. Discussion

**Role of weight decay.** One may argue that optimization biases, such as weight decay, are the primary drivers of generalization, with intrinsic symmetries playing only a secondary role. To examine this possibility, we conduct an additional experiment on modular addition without weight decay. As shown in Figure 8, even in the absence of explicit regularization, the model reliably converges to perfect test accuracy, and generalization emerges more rapidly when explicit symmetries are present. These results indicate that weight decay is not a necessary condition for generalization in this setting, and that intrinsic task symmetries alone can suffice to drive symmetry acquisition and generalization. We further argue that regularization plays a supplementary role, primarily stabilizing generalization during the geometric organization stage. We note that such observation is consistent with previous works (Kumar et al., 2024; Prieto et al., 2025).

**Comparison with prior work** A closely related line of work is due to Nanda et al. (2023), which decompose grokking into memorization, circuit formation, and cleanup. In contrast, our work addresses a complementary question: *why* generalization emerges in the first place. Rather than reverse-engineering a task-specific circuit, we identify intrinsic task symmetries as a general driver that shapes representation geometry across diverse algorithmic tasks. While Nanda et al. (2023) attributes circuit formation due to weight decay selecting a low-norm solution, we argue that symmetry acquisition is driven by intrinsic task structure. From this perspective, weight decay acts to stabilize and simplify representations that have already been aligned through symmetry acquisition, rather than serving as the primary force that induces generalizing structure.

## 7. Symmetry-driven geometry and generalization

We argue that intrinsic task symmetries help explain the geometric organization of learned representations. In structured algorithmic tasks, data are governed by algebraic or relational rules such as commutativity, associativity, distance constraints, or transitivity. When such symmetries are learned under a low-dimensional representation bias, embeddings tend to organize into structured geometric forms.

This view is related to the manifold hypothesis (Fefferman et al., 2016): although the ambient representation space may be high-dimensional, the relevant structure often lies on a low-dimensional manifold. In our setting, this structure is induced by the rules of the task. Thus, we interpret the observed geometries as arising from the interaction between task symmetry, partial memorization, and low-dimensional embedding constraints.

For modular arithmetic, commutativity and associativity lead to an abelian group structure, which explains the emergence of closed helical embeddings observed in Section 4. We formalize this with proofs deferred to Appendix J. We

also give partial explanations for graph metric completion and comparison tasks, where distance constraints and transitivity induce graph-like or grid-like structures.

Finally, Section 7.3 establishes a reconstruction result for cyclic groups: once the cyclic group structure is assumed, a small subset of Cayley table entries determines the full operation. This gives a concrete mechanism by which symmetry enables generalization from partial observations, within a highly structured algebraic setting.

Overall, this section suggests that the observed geometric organization is not merely a visualization artifact, but can arise from symmetry, partial memorization, and low-dimensional representation bias. Optimization biases such as weight decay or implicit regularization (Barrett & Dherin, 2021) may further promote this organization by reducing the effective dimension of the learned representation. The proofs of propositions are presented in Appendix J.

## 7.1. Modular arithmetic

In modular addition task, we formally verify that the symmetries induce the structured embedding geometry. We consider $\mathcal{X} = \mathbb{Z}_p$ is continuously embedded into the interval connecting endpoints : $\mathbb{Z}_p \subset \mathcal{M} = [0, p]/\{0 \sim p\}$. Note that the modular addition in $\mathbb{Z}_p$ is preserved in $\mathcal{M}$. If $\mathcal{M}$ has certain geometry, $\mathbb{Z}_p$ also follows the geometry. If $\mathcal{M}$ has commutativity and associativity, we can say that $\mathcal{M}$ has abelian group structure and $\mathcal{M}$ should be isomorphic to the closed helix in high-dimensional torus [2].

**Proposition 7.1.** *Let $\mathcal{M}$ be an one-dimensional compact abelian topological group and $\mathbb{Z}_p$ is continuously embedded in $\mathcal{M}$. Then $\mathcal{M}$ is isomorphic to $S^1$ embedded in high-dimensional torus $\mathbb{T}^D$ (closed helix)*

$$\Phi : \mathcal{M} \cong S^1 \hookrightarrow \mathbb{T}^D.$$

*Hence $\Phi(\mathbb{Z}_p) \hookrightarrow \Phi(\mathcal{M})$ is also confined in the closed helix.*

Using Proposition 7.1, since $\mathcal{M}$ is compact abelian group (satisfying commutativity and associativity), $\mathcal{M}$ is isomorphic to the closed helix. Here, *isomorphism* refers to a structure-preserving bijection (homeomorphism in topological sense). Consequently, we theoretically observe that if the network sufficiently learns the commutativity and associativity of modular addition, its embeddings naturally organize along a closed helical structure in the representation space. This observation further suggests that other modular operations exhibiting similar symmetries give rise to analogous geometric structures (see Appendix I).

---

[2]By the definition of the abelian group, $\mathcal{M}$ should have identity and invertible conditions to be an abelian group. Such properties are simple and easy for the network to learn, so we skip the conditions.

## 7.2. Graph metric completion and comparison tasks

We also provide a partial theoretical analysis of the geometric organization observed in graph metric and comparison tasks. When the embedding space is assumed to be $\mathbb{R}^d$ or $\mathbb{T}^d$ for small $d \in \mathbb{N}$, it is possible to obtain partial explanations for the emergence of structured geometry.

Let $\mathcal{X} = \{v_1, \ldots, v_n\}$ be a set of vertices of a graph with an embedding $\Phi : \mathcal{X} \to \mathbb{R}$ into a low-dimensional space. If $\Phi(\cdot)$ incorporates partial distance information (memorization of training data), intrinsic symmetries within the task, and low embedding dimension, then the image $\Phi(\mathcal{X})$ exhibits an organized geometric structure. The following proposition formalizes this intuition for the case of Path or Cycle graphs (Figure 3). If $\mathcal{X}$ is embedded into $\mathbb{R}$ preserving triangular symmetry, the resulting embedding recovers the graph structure.

**Proposition 7.2.** *Let $\mathcal{X} = \{v_1, \ldots, v_n\}$ be a one-dimensional Path or Cycle graph (see Figure 3) with $n$ vertices ($n \geq 3$) and $\mathcal{Z} \in \{\mathbb{R}, \mathbb{T}\}$ be a embedding space. Let $\Phi : G \to \mathcal{Z}$ be embedding with partial distance information*

$$d(\Phi(v_i), \Phi(v_{i+1})) = 1, \ i = 1, \ldots n - 1,$$
$$d(\Phi(v_1), \Phi(v_n)) = 1, \ \ \text{if } \mathcal{X} \text{ is Cycle}.$$

*If $\Phi(\cdot)$ preserve the triangular symmetry, then $\Phi(\mathcal{X})$ has its graph structure in embedding space $\mathbb{R}$.*

Since other graphs in Figure 3 are considered as a composition of Path and Cycle, we can expect similar embedding structure.

An analogous result holds for the comparison task. The following proposition states that if $\mathcal{X}$ is embedded into $\mathbb{R}$ while preserving transitivity, the resulting embedding exhibits a grid structure.

**Proposition 7.3.** *Let $\mathcal{X} = \{v_1, \ldots, v_n\}$ be a set of nodes in 1D comparison task (see Section 3.3) with $n \geq 3$ nodes. Let $\Phi : \mathcal{X} \to \mathbb{R}$ be embedding with partial relation information*

$$\Phi(v_i) < \Phi(v_{i+1}), \ i = 1, \ldots n - 1.$$

*If $\Phi(\cdot)$ preserve the transitivity, then $\Phi(\mathcal{X})$ has a grid structure in embedding space $\mathbb{R}$.*

Taken together, these observations suggest that geometric organization can be explained through the interplay of intrinsic task symmetries, sufficient memorization of partial information, and low-dimensional embedding constraints.

## 7.3. Symmetry-driven generalization

We note that for structured tasks such as modular addition and multiplication, one can explicitly establish a theoretical connection between symmetry and generalization. In

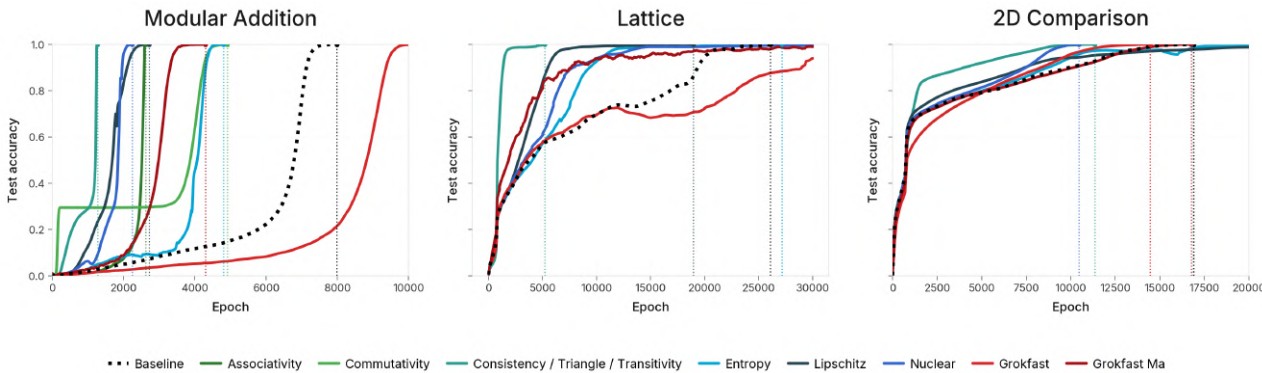

*Figure 9.* Comparison of convergence speed on modular arithmetic tasks (x+y), graph metric completion (lattice), and comparsion (2D) task with **Geometry-promoting priors** (*nuclear*, *entropy*, *lipschitz*); **Symmetry-prompting losses** (*commutativity/associativity/*etc); **Optimizer-based accelerants** (*GrokFast*). Markers denote the first point where accuracy reaches 1. *Consistency* refers to using both Commutativity and Associativity

particular, symmetry constraints (e.g., commutativity and associativity), together with partial memorization, can suffice to reconstruct the entire operation. This provides a concrete example in which symmetry enables generalization from only a small subset of observed input pairs.

In particular, under the assumption that the operation forms a cyclic group, we show that only a fraction of the Cayley table is required to recover the entire operation.

**Proposition 7.4.** *Let $G = \{0, 1, \ldots, p-1\}$ and let $\odot$ be a binary operation on $G$. Assume that $(G, \odot)$ is a cyclic group of order $p$, with identity element $0$ and generator $1$. Then there exists a subset $M \subset G \times G$, with $|M| = 2p - 1$, such that the values of $\odot$ on $M$, namely $\tilde{M} := \{(a, b, c) \in G^3 : (a, b) \in M, \ a \odot b = c\}$, uniquely determine the full Cayley table of $(G, \odot)$. Consequently, the fraction of entries required to reconstruct the operation is*

$$\frac{|M|}{|G \times G|} = \frac{2p-1}{p^2} \sim \frac{2}{p}.$$

# 8. Acceleration method for fast generalization

## 8.1. Setup

While previous observations indicate a strong correlation between symmetry and generalization, here we validate the causal link between them. Specifically, we examine whether explicitly promoting symmetry can induce or accelerate the onset of generalization. To this end, we introduce two distinct auxiliary loss functions. The total training objective is defined as a weighted sum of the cross-entropy loss, weight decay, and the proposed auxiliary term:

$$\mathcal{L}_{\text{total}} = \mathcal{L}_{\text{CE}} + \alpha\|\theta\|^2 + \lambda\mathcal{L}_{\text{aux}}$$

where $\alpha$ and $\lambda$ are hyperparameters. For full configuration details, refer to Appendix G.

## 8.2. Symmetry-prompting loss

Based on the intuition developed in Section 6, we propose a symmetry-prompting loss designed to accelerate generalization. For modular arithmetic tasks, we use **commutativity** and **associativity** of addition and multiplication, along with suitable symmetric proxy losses (see Table 1). For graph metric completion and comparison tasks, we instead leverage **triangular symmetry** and **transitivity**.

## 8.3. Geometry-prompting priors

Based on the intuition developed in Section 7, we observe that generalization stabilizes during the geometric organization stage. Accordingly, to further promote generalization, we propose geometry-prompting priors that encourage low-rank structure. Specifically, we consider nuclear norm regularization, which minimizes the sum of singular values; entropy regularization, which discourages diffuse, high-energy embeddings; and Lipschitz regularization, which enforces local smoothness. This intuition is consistent with prior findings in Notsawo et al. (2025), which employ nuclear norm regularization, and in DeMoss et al. (2025), which penalize entropy.

The explicit loss formulations and implementation details for each variant are provided in Appendix E.

## 8.4. Results

We report experimental results for the proposed acceleration methods. As shown in Figure 9, symmetry-prompting strategies consistently reduce generalization time relative to the baseline and prior acceleration methods across all tasks. Geometry-prompting alone also improves generalization speed compared to the baseline, consistent with recent findings that geometric structure can facilitate grokking (Walker et al., 2025; DeMoss et al., 2025; Notsawo et al., 2025).

However, these gains are systematically unstable than those achieved through symmetry prompting.

In contrast, while Grokfast (Lee et al., 2024) accelerates grokking in some regimes, we find that it can slow training in others, indicating sensitivity to task structure. Overall, these results support the view that explicit alignment with intrinsic task symmetries provides a more reliable and causal mechanism for accelerating generalization than geometry alone or optimization-level heuristics. Additional experimental results are reported in Appendix E.

We further verify that these effects are not specific to Transformer architectures. As illustrated in Figure 8, symmetry prompting also accelerates generalization in MLPs and remains effective even in the absence of weight decay, reducing the time to generalization by up to two orders of magnitude. Finally, to further assess the causal role of symmetry, we conduct a complementary intervention that suppresses symmetry acquisition, akin to Han et al. (2026). As reported in Appendix F, penalizing symmetry acquisition impairs generalization, further supporting the role of symmetry as a driver of grokking.

## 9. Conclusion

In this paper, we establish that intrinsic task symmetry is a primary driver of generalization in algorithmic reasoning. By analyzing modular arithmetic, graph metric completion, and comparison tasks, we show that generalization emerges when models move beyond memorization to learn the intrinsic symmetries of the target tasks. Our results suggest a consistent three stage dynamic of memorization, symmetry acquisition, and geometric organization, where the shift to perfect test accuracy specifically coincides with the alignment of representations to task-intrinsic symmetries. We provide theoretical and empirical evidence that these symmetries, combined with low rank constraints, steer embeddings into structured geometries.

Furthermore, we demonstrate the causal role of symmetry through symmetry- and geometry-prompting strategies that significantly accelerate the grokking process. By explicitly regularizing with symmetry-prompting loss, we reduce the time to generalization across diverse algorithmic domains. These findings identify symmetry as the mechanism that gives rise to the specific shape of learned representations and provide a principled explanation for why structured geometries emerge after generalization. Our work establishes a unified view of grokking as a process of symmetry acquisition that is essential for robust algorithmic reasoning.

## Limitations

While our results identify intrinsic task symmetry as a unifying mechanism across the algorithmic domains studied in this work, the scope of this conclusion is subject to several limitations. First, our experiments are conducted in controlled algorithmic settings where the relevant symmetries are either known or can be explicitly specified. This design allows us to isolate the role of symmetry in generalization, but it does not by itself establish that the same symmetry-acquisition dynamics will transfer unchanged to large-scale language modeling or to real-world datasets, where the governing invariances may be unknown, approximate, or difficult to disentangle.

Also, although our acceleration experiments provide interventional evidence that symmetry acquisition plays an important role in generalization, the proposed symmetry-prompting objectives rely on access to candidate task symmetries. When such symmetries are not known a priori, directly applying these objectives may be difficult. Our geometry-prompting results suggest a possible alternative route; and we also provide an initial step toward automatically identifying relevant symmetry structure in Appendix H; nevertheless, developing scalable methods for symmetry discovery in more complex settings remains an important direction for future work.

## Acknowledgement

This work was supported by the National Research Foundation of Korea (NRF) grant (RS-2026-25472534).

## Impact Statement

This paper presents work whose primary goal is to advance the theoretical and empirical understanding of neural network generalization in algorithmic reasoning tasks. We do not anticipate any direct negative societal consequences or deployment-specific risks arising from this research. More broadly, insights from this work may contribute to the development of more reliable and interpretable learning systems.

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

# A. Training Dynamics of Grokking

In this section, we present the figure for complete training dynamics of each task conducted in our experiments, visualizing the distinct three-stage progression: memorization, symmetry acquisition, and geometric organization.

## A.1. Modular Arithmetic Tasks

For modular arithmetic, we test **Commutativity and Associativity** as our symmetric criterion, which can be computed from the model's output logits by comparing predictions along algebraically equivalent computation paths. For operations that do not naively satisfy these properties (e.g., subtraction), we employ a suitable *proxy identity*, which is detailed in Table 1.

**Notation.** We denote by $p_\theta(\cdot \mid a, b)$ the model's predicted distribution (softmax over logits) given inputs $(a, b)$, restricted to $\{0, \ldots, p-1\}$. To quantify similarity between two distributions $p$ and $q$, we use the *symmetric KL divergence*:

$$\mathrm{KL}_{\mathrm{sym}}(p, q) \;=\; \tfrac{1}{2}\Big(\mathrm{KL}(p \parallel q) + \mathrm{KL}(q \parallel p)\Big).$$

### A.1.1. ADDITION: $x + y \pmod{p}$

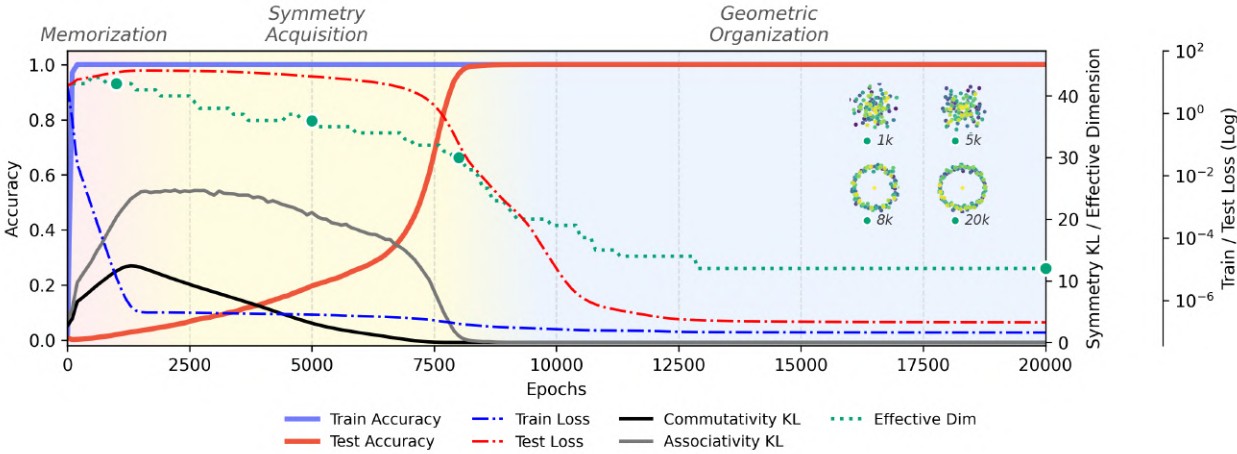

*Figure 10.* **Grokking Dynamics in Modular Addition.**

### A.1.2. SUBTRACTION: $x - y \pmod{p}$

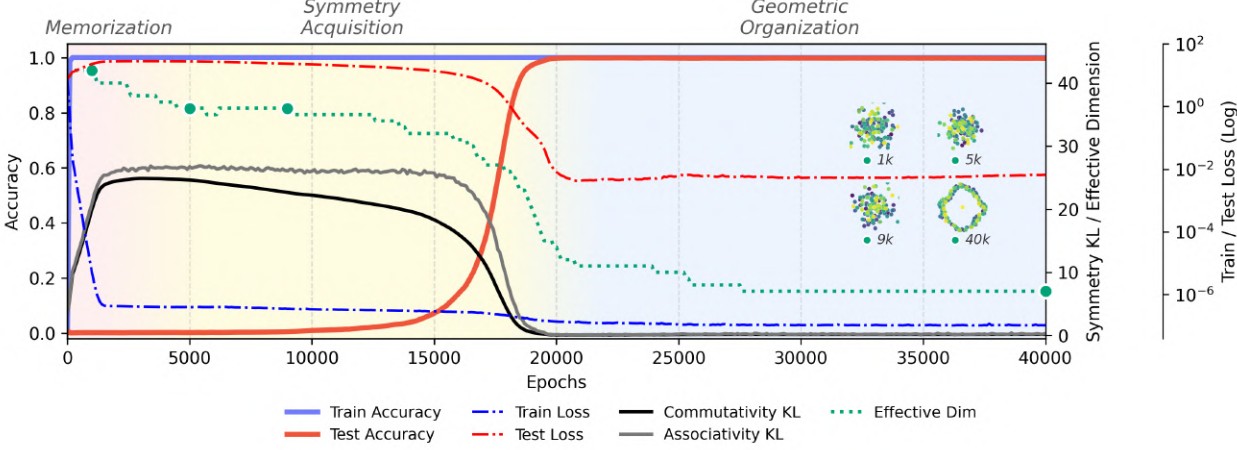

*Figure 11.* **Grokking Dynamics in Modular Subtraction.**

### A.1.3. SQUARED SUM: $(x+y)^2 \pmod{p}$

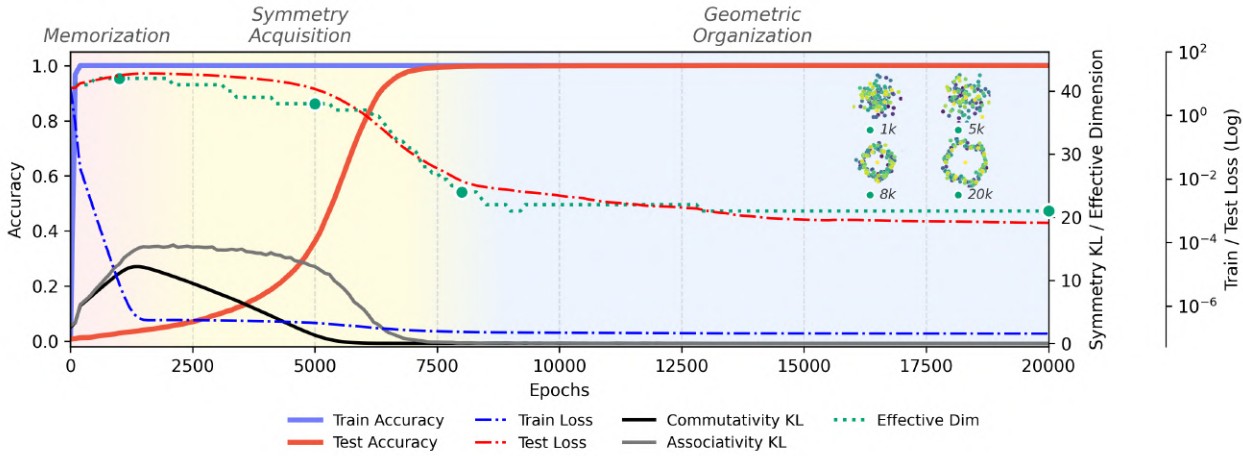

*Figure 12.* **Grokking Dynamics in Modular Squared Sum**

### A.1.4. CUBED SUM: $(x+y)^3 \pmod{p}$

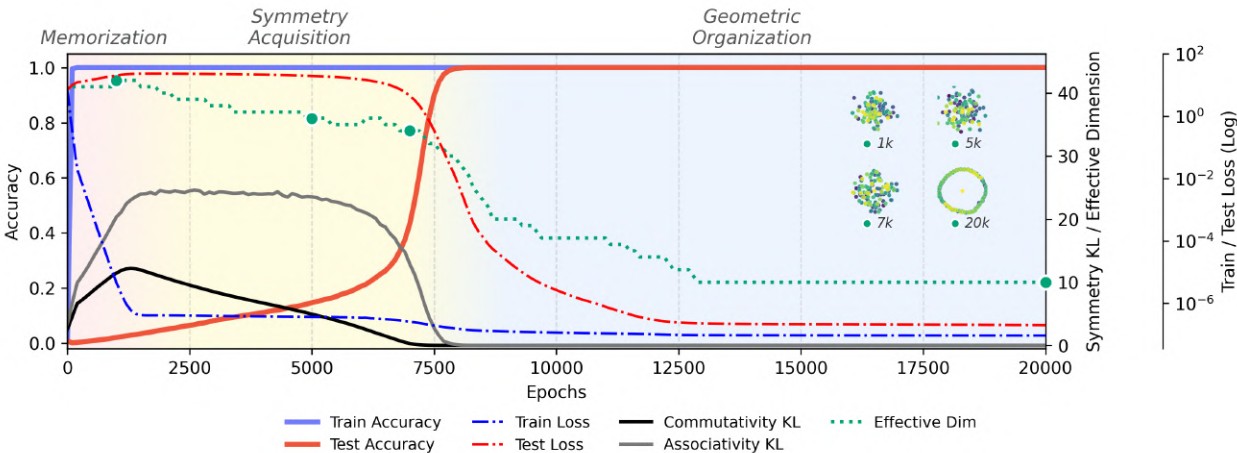

*Figure 13.* **Grokking Dynamics in Modular Cubed Sum**

## A.1.5. MULTIPLICATION: $x \times y \pmod{p}$

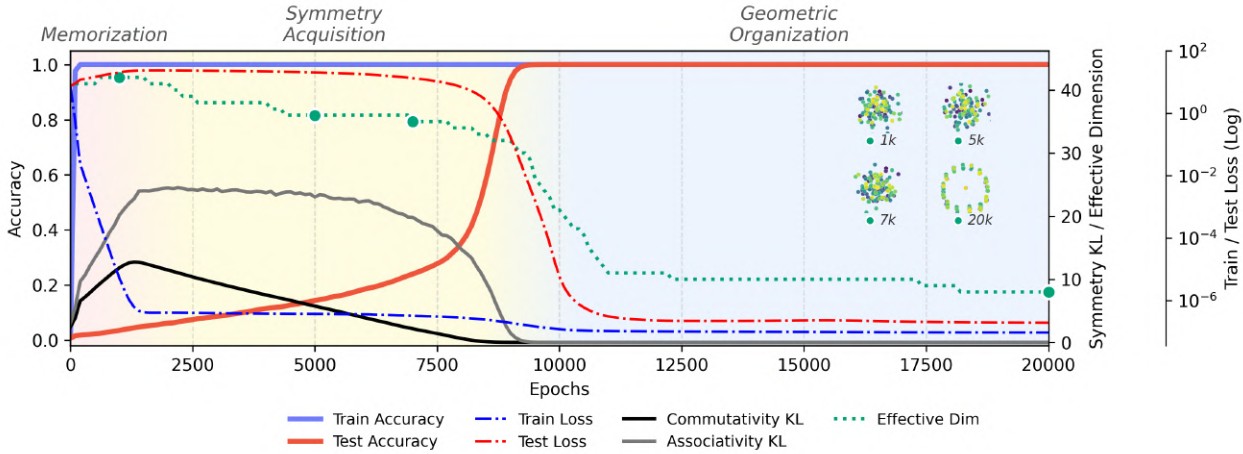

*Figure 14.* **Grokking Dynamics in Modular Multiplication**

## A.1.6. AFFINE–BILINEAR: $x + xy + y$

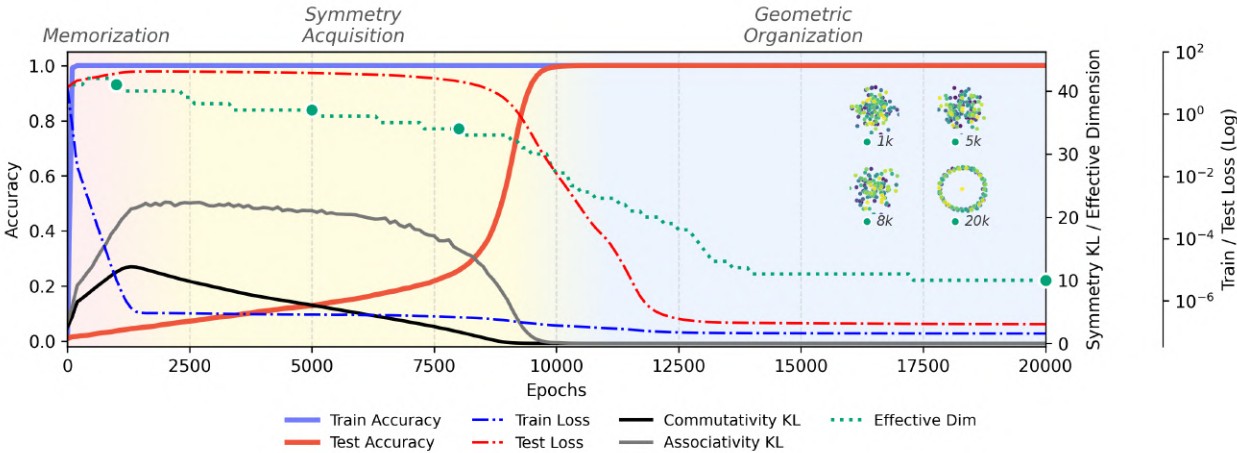

*Figure 15.* **Grokking Dynamics in Modular Affine–bilinear**

### A.2. Geometric Progress Measure for Graph Metric Completion Tasks

For the graph metric completion task, we use the **Triangle Equality** property. This measure evaluates whether the model's learned representation respects the additive nature of distances along shortest paths. Formally, for any three nodes $u, v, w$, if $v$ lies on a geodesic (shortest path) between $u$ and $w$, the metric must satisfy: $d(u, w) = d(u, v) + d(v, w)$

To quantify this, we sample triplets $(a, b, c)$ such that $b$ is an intermediate node on a shortest path between $a$ and $c$. Then, we evaluate the consistency of the model's predictions by computing the convolution of the predicted probability distributions for $d(a, b)$ and $d(b, c)$, denoted as $P_{a \to b} * P_{b \to c}$. We then measure the Symmetric KL Divergence between this convolved distribution and the model's direct prediction for $d(a, c)$: $\mathcal{L}_{\text{tri}} = D_{\text{KL}} \left( (P_{a \to b} * P_{b \to c}) \parallel P_{a \to c} \right)$.

#### A.2.1. CYCLE

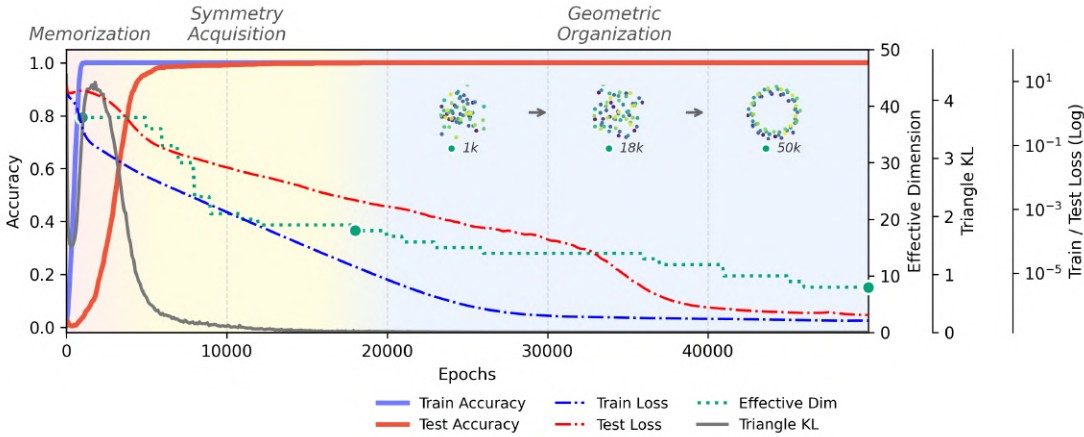

*Figure 16.* Convergence speed on **cycle** graph metric completion when using various regularization.

#### A.2.2. PATH

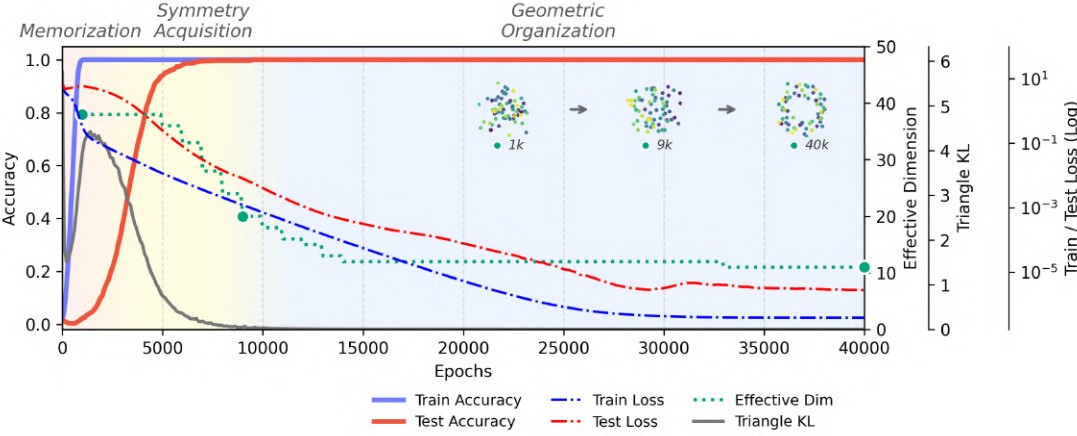

*Figure 17.* Convergence speed on **path** graph metric completion when using various regularization.

### A.2.3. CYLINDER

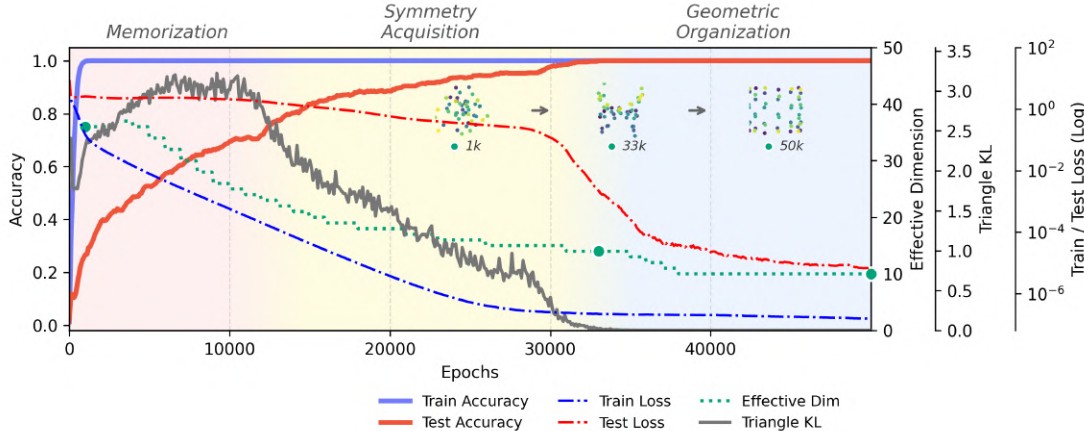

*Figure 18.* Convergence speed on **cylinder** graph metric completion when using various regularization.

### A.2.4. HYPERCUBE

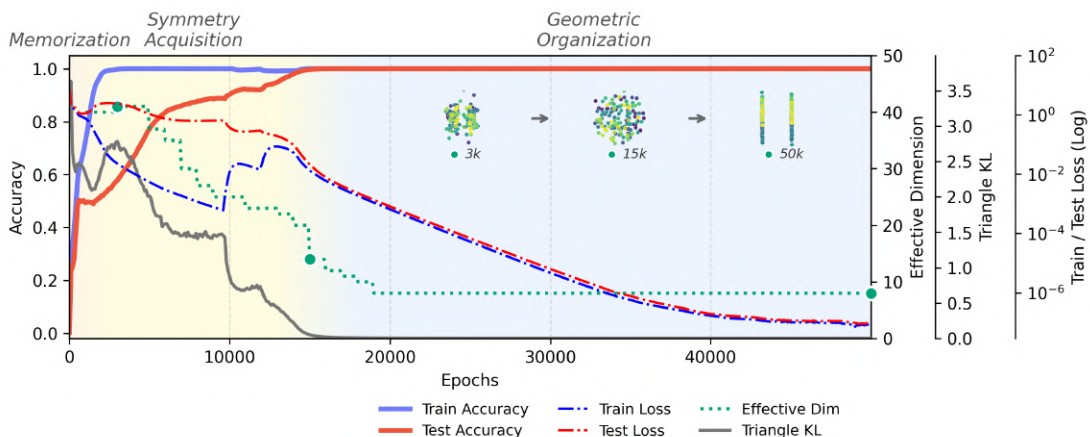

*Figure 19.* Convergence speed on **hypercube** graph metric completion when using various regularization.

## A.2.5. 2D LATTICE

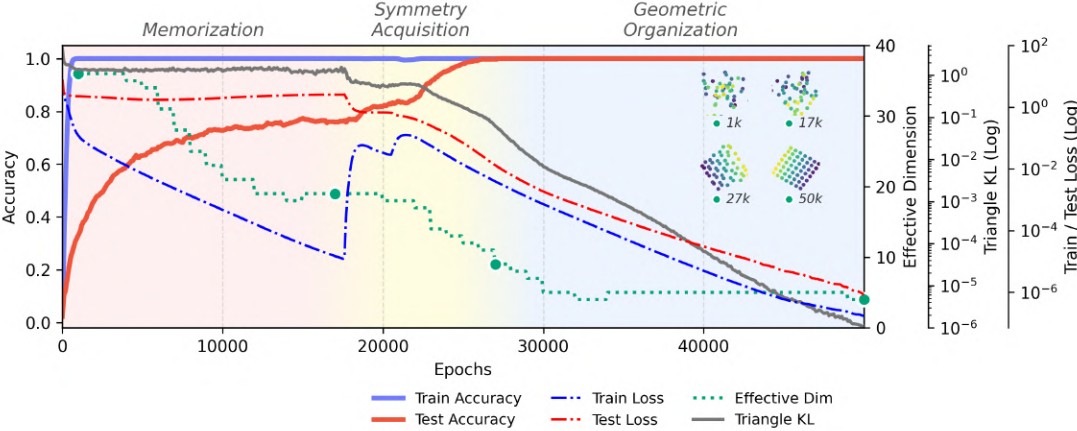

*Figure 20.* Convergence speed on **2d lattice** graph metric completion when using various regularization.

## A.2.6. 3D LATTICE

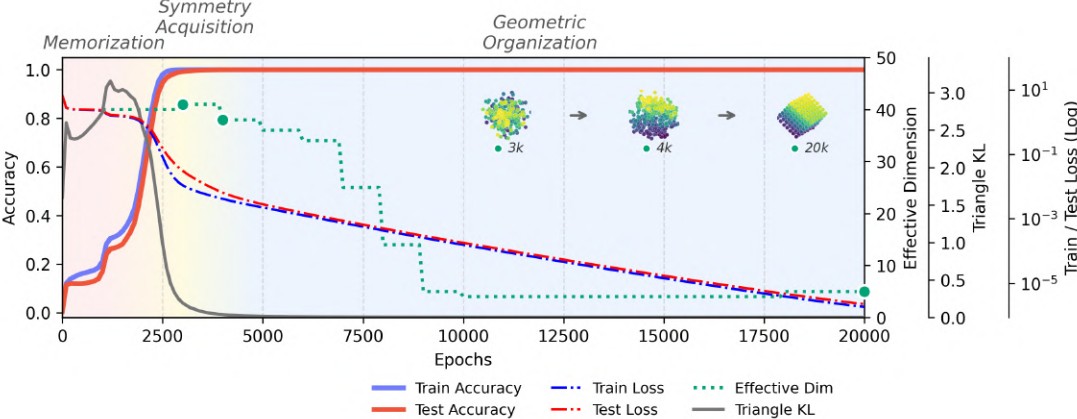

*Figure 21.* Convergence speed on **3d lattice** graph metric completion when using various regularization.

### A.3. Comparison Tasks

For the comparison task, we use the **Transitivity** property. We evaluate whether the model's pairwise comparisons are consistent with a global linear ordering, specifically testing if the learned representations adhere to a Bradley-Terry model, *i.e.* the probability $P(a > b)$ is determined by the difference in latent scores, implying that the log-odds should be additive: $\ell_{ac} \approx \ell_{ab} + \ell_{bc}$.

In practice, we sample triples $(a, b, c)$ strictly ordered by their ground-truth rank such that $a < b < c$. We first extract the model's binary log-odds for each pair by taking the difference between the *greater* and *less* logits: $\ell = \text{logit}_{>} - \text{logit}_{<}$. After, we compare two distributions: the model's direct prediction $P(a > c) = \sigma(\ell_{ac})$ and the transitive prediction implied by the intermediate steps $Q(a > c) = \sigma(\ell_{ab} + \ell_{bc})$. The final metric is the Symmetric KL Divergence between these two distributions: $\mathcal{L}_{\text{trans}} = \frac{1}{2} \left( D_{\text{KL}}(P \parallel Q) + D_{\text{KL}}(Q \parallel P) \right)$.

### A.3.1. 2D COMPARISON

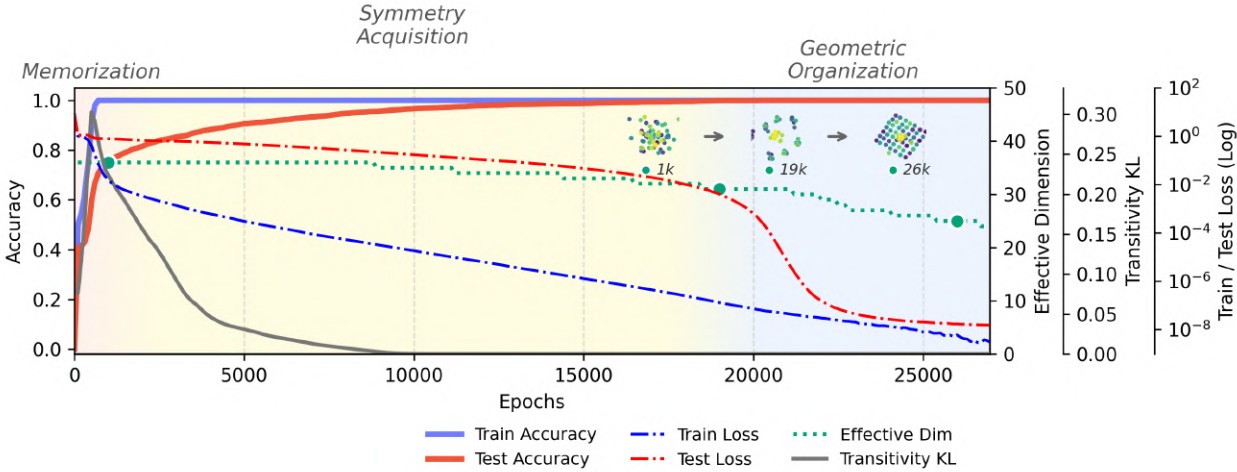

*Figure 22.* **Grokking Dynamics in 2D Comparison**

### A.3.2. 3D COMPARISON

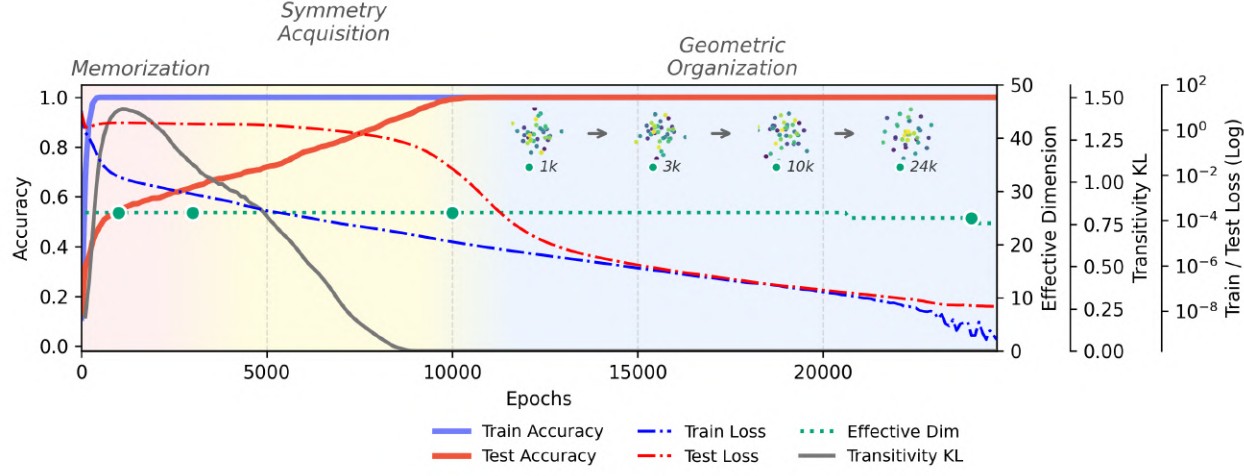

*Figure 23.* **Grokking Dynamics in 3D Comparison**

# B. Gradient Analysis

To connect our findings to training dynamics, we analyze gradient alignment during training. At each logged step, we first compute the pure autograd gradients of individual component losses, including the training loss, symmetry losses, and weight decay term, where the symmetry losses are estimated from randomly sampled symmetry pairs or triplets. We then perform the actual training update and compare each component gradient with the realized optimizer direction, defined as $\theta_t - \theta_{t+1}$. Thus, the measured cosine similarity reflects alignment with the full optimizer-induced update, including AdamW moments, learning-rate scaling, decoupled weight decay, and other optimizer effects.

We observe that during the symmetry acquisition phase, the optimizer direction is positively aligned with the gradients of the symmetry constraints, suggesting that training implicitly follows directions that reduce symmetry violations and thereby promote generalization. Since the instantaneous cosine estimates display high variance, we report smoothed curves using a moving average for visualization.

## B.1. Modular Addition

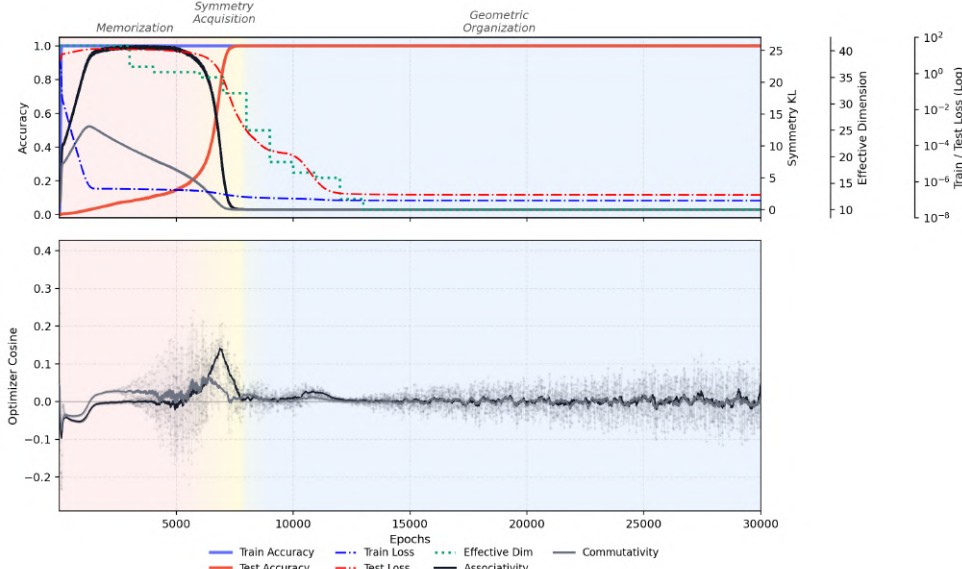

*Figure 24.* Gradient analysis of ***commutativity*** and ***associativity*** on the modular addition task.

## B.2. Lattice

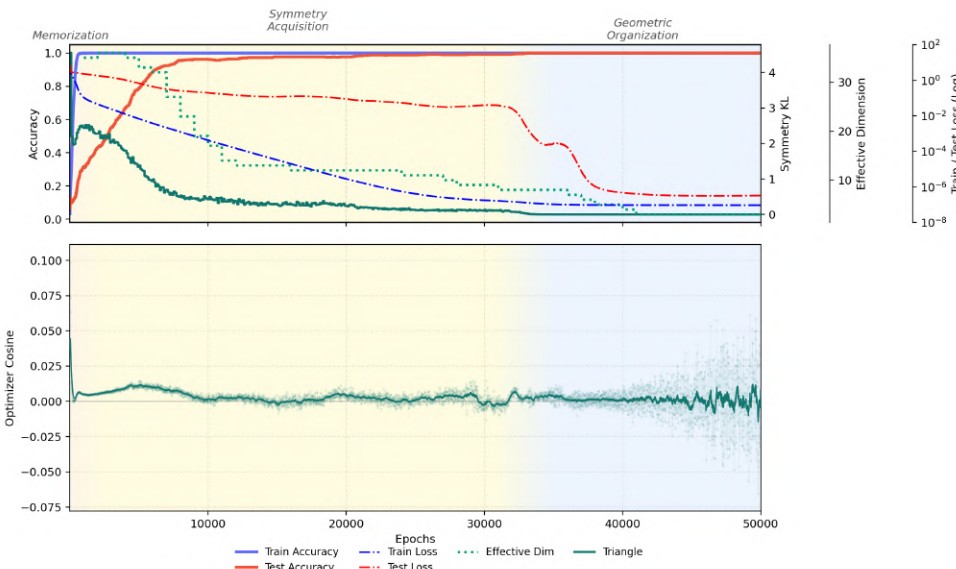

*Figure 25.* Gradient Analysis of **Triangle Symmetry** on Lattice, Graph Completion Task.

## B.3. 2D Comparison

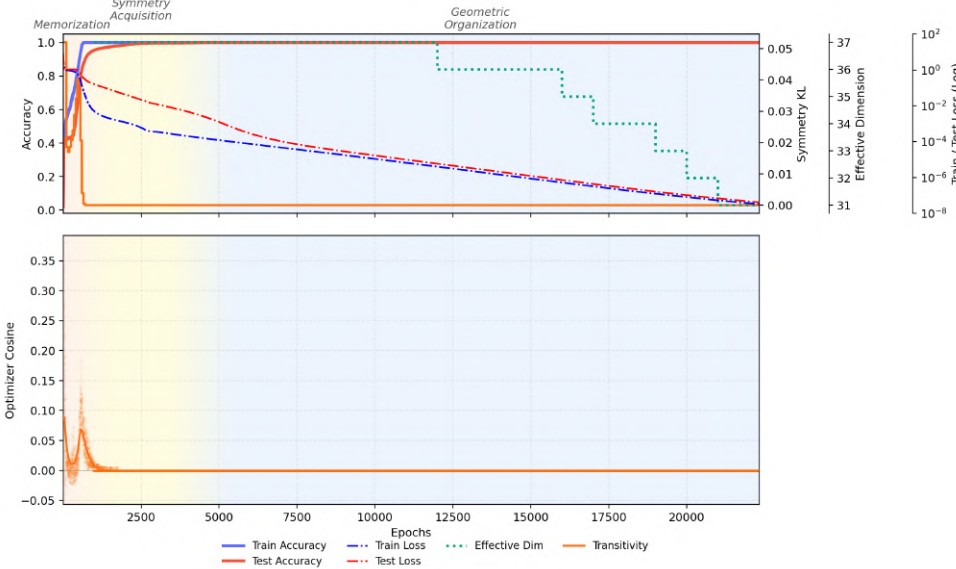

*Figure 26.* Gradient Analysis of **Transitivity Symmetry** on 2D Comparison Task.

## C. Characterizing Generalization via Intrinsic Symmetry

Below we plot the maximum *instance-level* symmetry criterion achieved during training for each task. We observe a high correlation between low maximum instance-level criterion and successful generalization. These results suggest that the elimination of structural outliers represented by the maximum symmetry violation is a key marker of the phase transition from memorization to true algorithmic understanding.

For each task, we track 20 independent seeds. In the scatter plots, each point represents the state of a run at a specific training epoch: **blue points** denote runs that have successfully generalized (reached 100% accuracy) by that epoch, while **red markers** indicate runs that have not yet generalized. We observe a *soft threshold* distinguishing these states, where generalizing runs consistently exhibit lower maximum symmetry violations compared to their non-generalizing counterparts.

### C.1. Modular Arithmetics

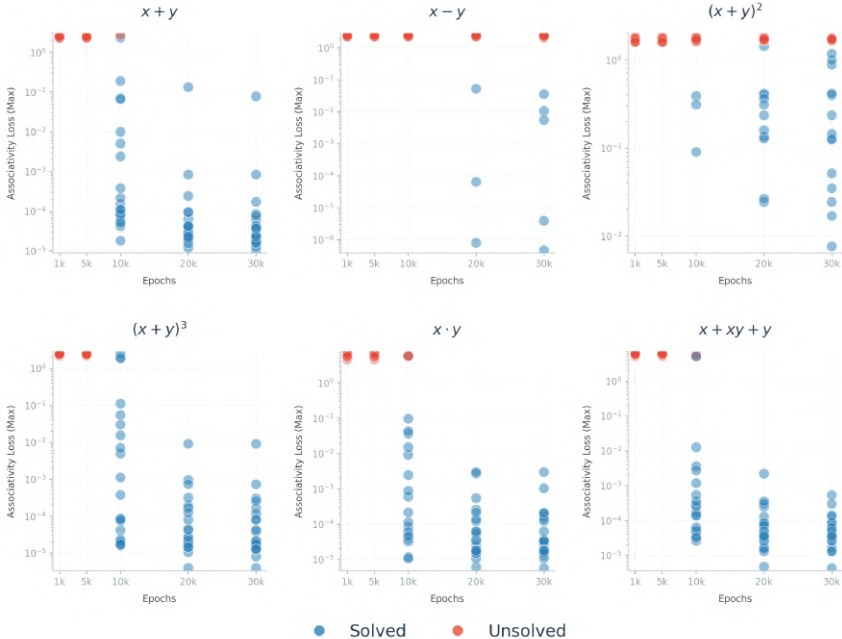

*Figure 27.* **Maximum Symmetry Violation and Generalization Status for** modular arithmetic **tasks.**

## C.2. Graph Metric Completion

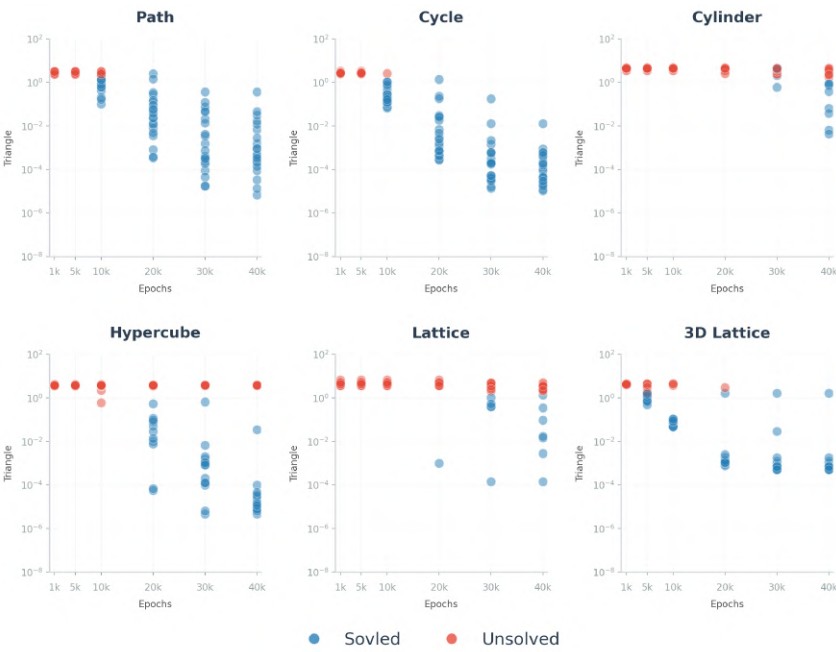

*Figure 28.* Symmetry Loss vs. Test Accuracy for Graph Metric Completion Tasks

## C.3. Comparison

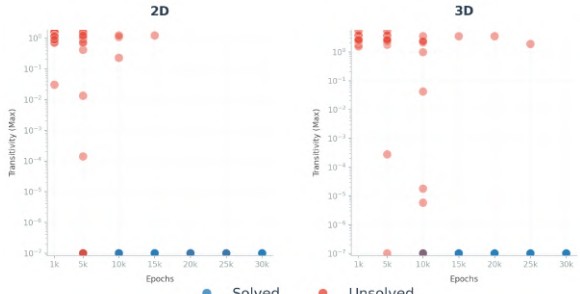

*Figure 29.* Symmetry Loss vs. Test Accuracy for Comparison Tasks

**Statistical Significance.** We additionally report the significance of the same soft-threshold test using $\mathrm{acc} > 0.99$ as the success criterion in Table 2. We use this relaxed criterion because exact $100\%$ accuracy can be reached substantially later in some runs due to a small number of residual errors, whereas $\mathrm{acc} > 0.99$ more directly captures the quantitative onset of the generalizing regime.

*Table 2.* Soft-threshold test $p$-values across 20 seeds with success defined as $\mathrm{acc} > 0.99$.

| Task | $p$-value |
|---|---|
| Modular Addition | $9.5367 \times 10^{-7}$ |
| Lattice | $2.734 \times 10^{-2}$ |
| Comparison (2D) | $3.125 \times 10^{-2}$ |

# D. Analysis of Embedding Geometry

## D.1. Effective Dimensionality

We quantify the effective dimensionality of the embedding space using a PCA-based criterion. We apply principal component analysis to the token embedding matrix and define the effective embedding dimension as the number of components whose explained variance exceeds 1%. This serves as a simple and robust proxy for the dimensionality of the learned representations.

## D.2. Full Visualization

Here, we visualize the geometric structure of the learned embeddings for each task using Principal Component Analysis (PCA). To fully capture the manifold's shape, we visualize the pairwise projections of the top principal components, plotting up to a maximum of 8 dimensions.

## D.3. Modular Arithmetic

### D.3.1. TASK 1 : ADDITION: $x + y \pmod{p}$

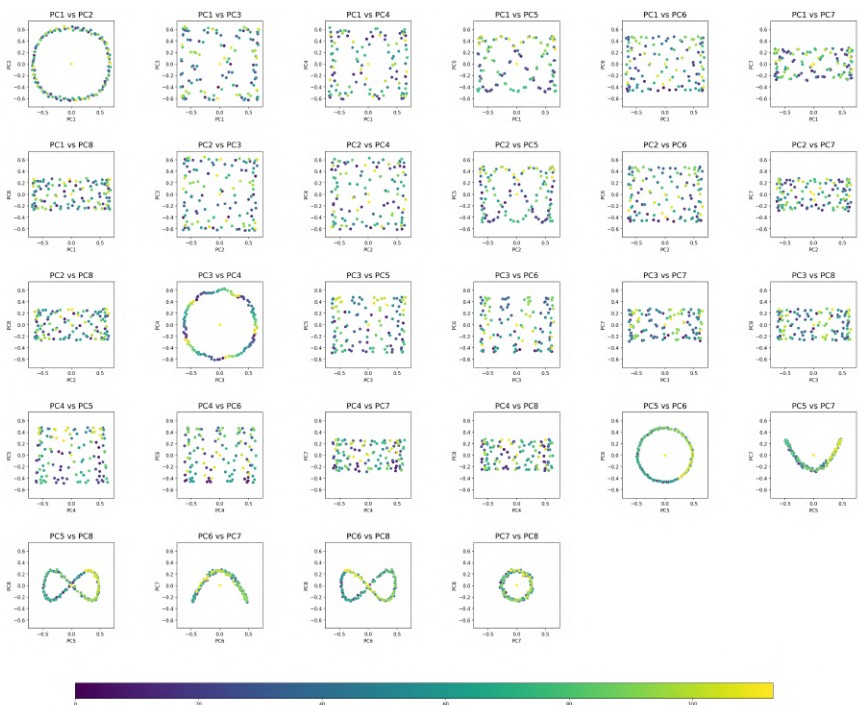

*Figure 30.* PCA projection of embeddings of model trained on modular addition task.

### D.3.2. TASK 2 : SUBTRACTION: $x - y \pmod{p}$

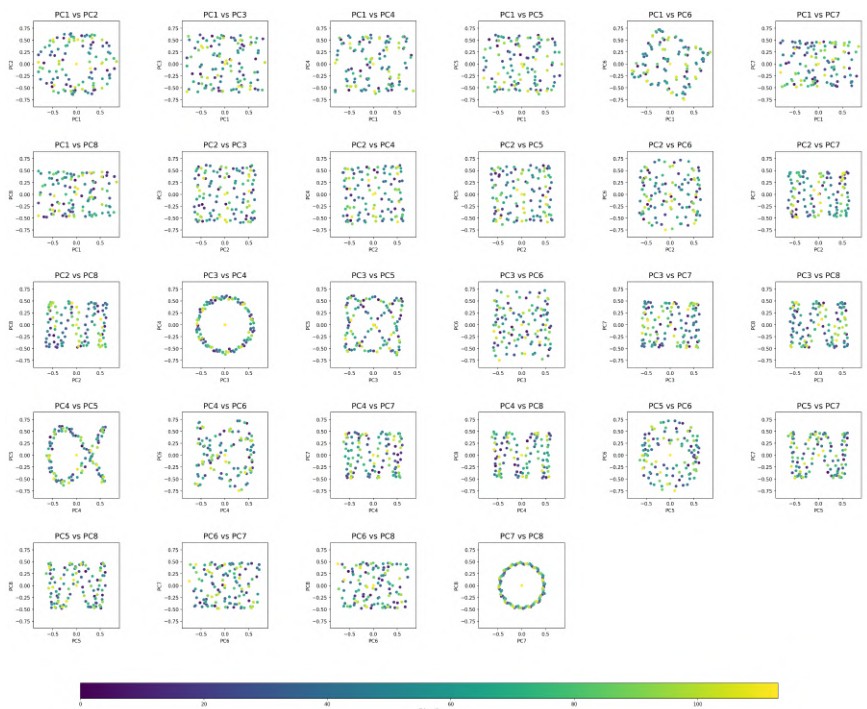

*Figure 31.* PCA projection of embeddings of model trained on modular subtraction task.

### D.3.3. TASK 3 : $(x + y)^2 \pmod{p}$

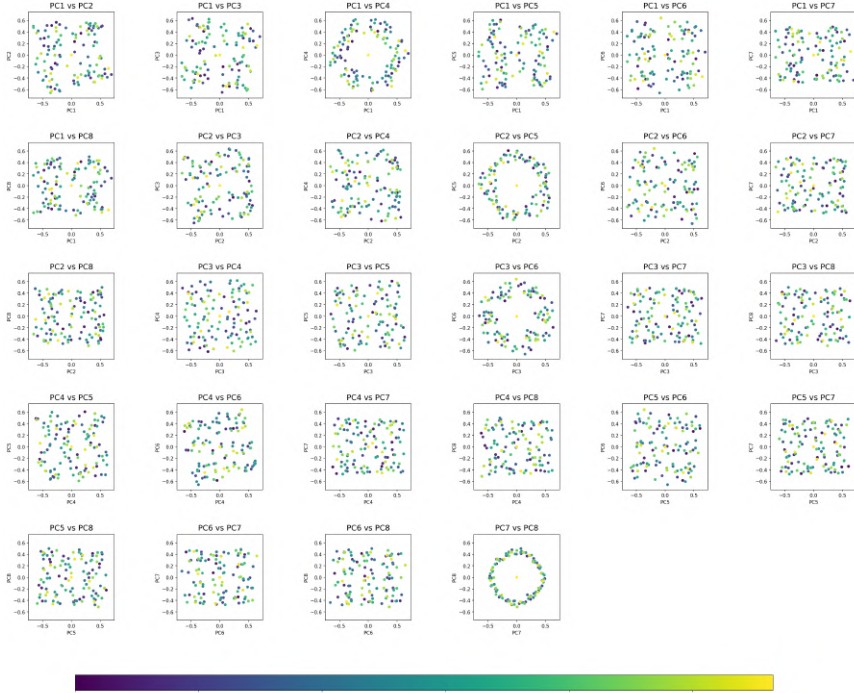

*Figure 32.* PCA projection of embeddings of model trained on Task 3.

### D.3.4. TASK 4 : $(x + y)^3 \pmod p$

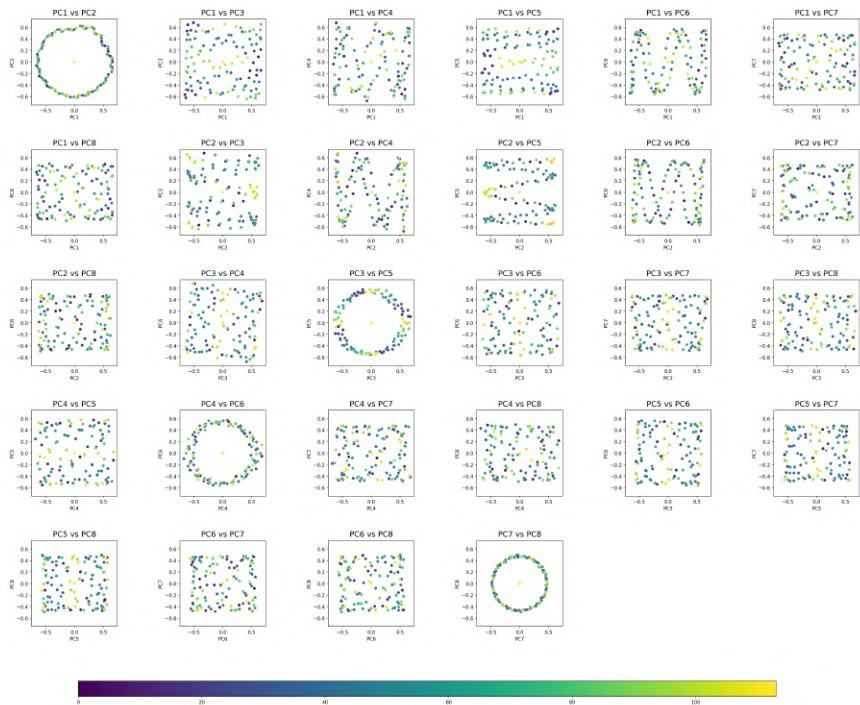

*Figure 33.* PCA projection of embeddings of model trained on Task 4.

### D.3.5. TASK 5 : MULTIPLICATION: $x \times y \pmod p$

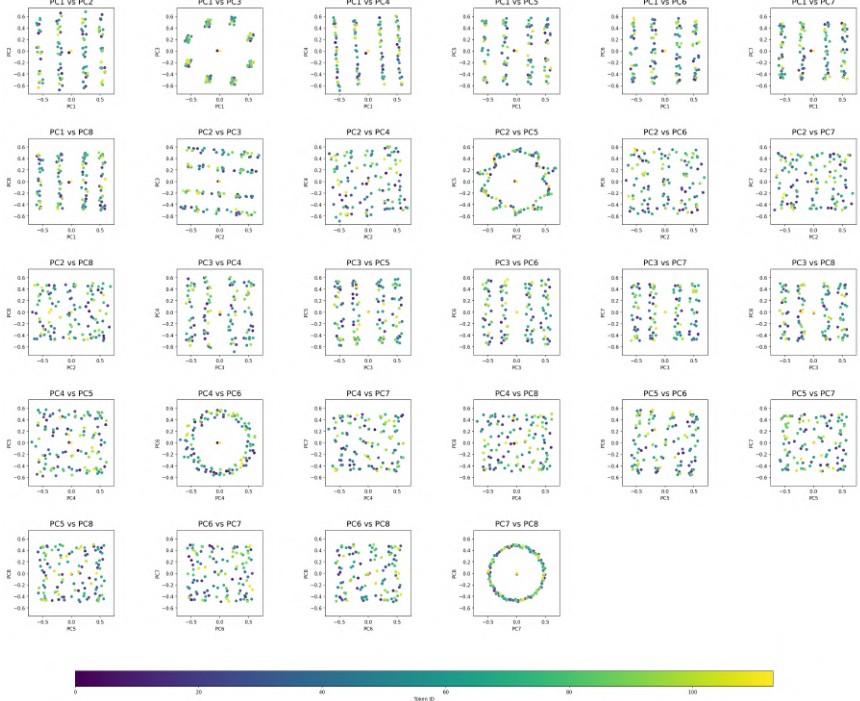

*Figure 34.* PCA projection of embeddings of model trained on modular multiplication task.

D.3.6. TASK 6 : $x + xy + y \pmod{p}$

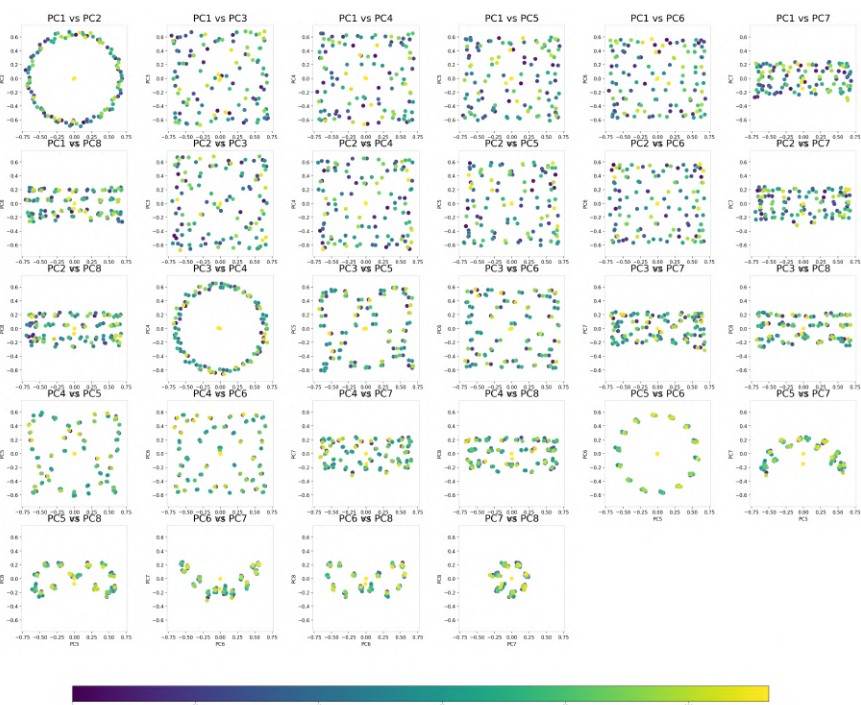

*Figure 35.* PCA projection of embeddings of model trained on Task 6.

## D.4. graph metric completion task

### D.4.1. PATH

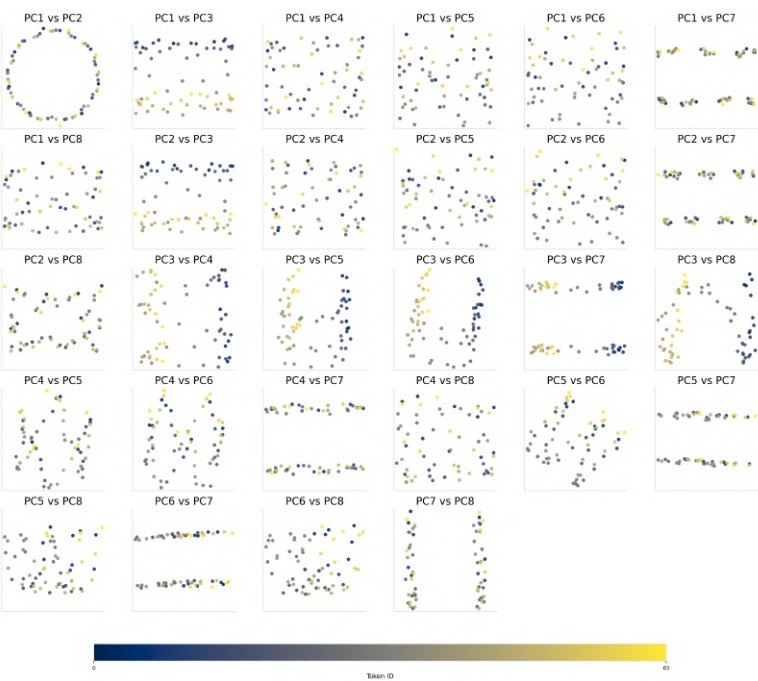

*Figure 36.* PCA projection of embeddings of model trained on the PATH graph metric completion.

### D.4.2. CYCLE

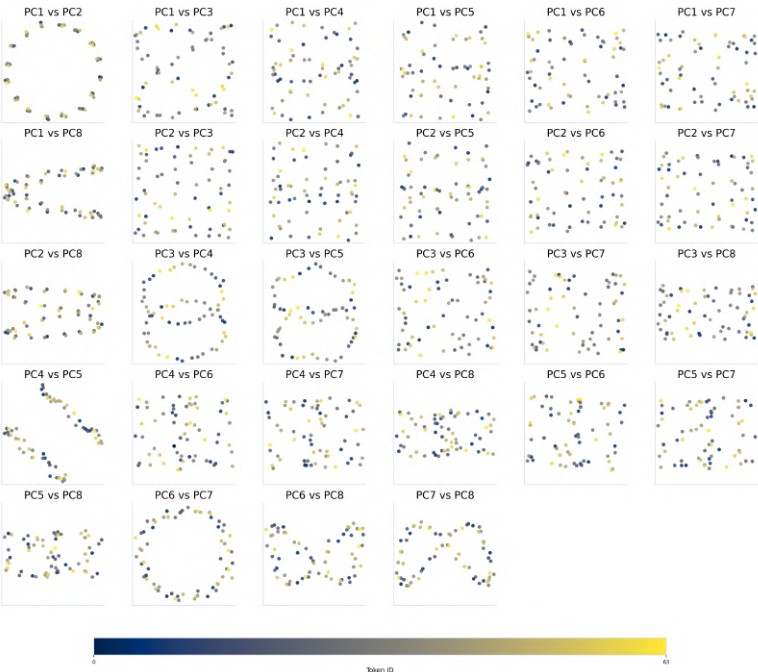

*Figure 37.* PCA projection of embeddings of model trained on the CYCLE graph metric completion.

### D.4.3. CYLINDER

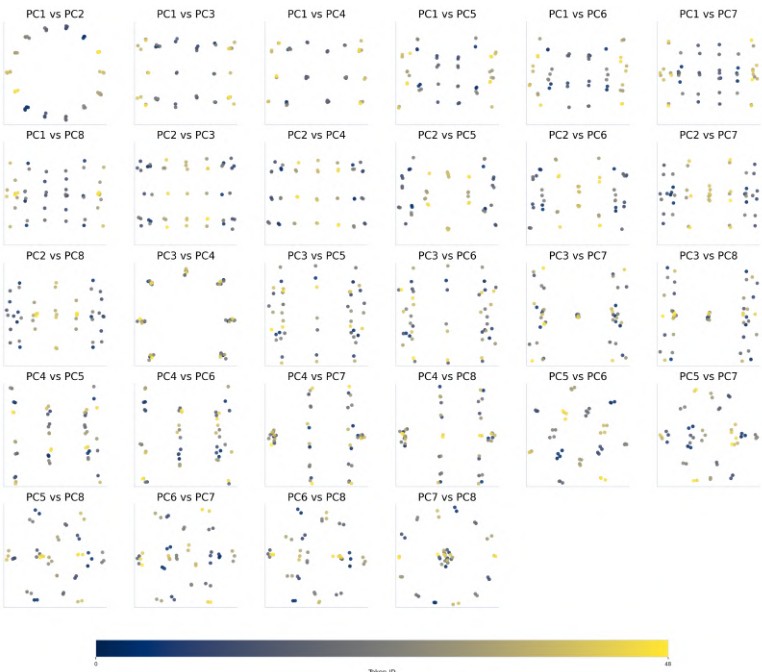

*Figure 38.* PCA projection of embeddings of model trained on the CYLINDER graph metric completion.

### D.4.4. HYPERCUBE

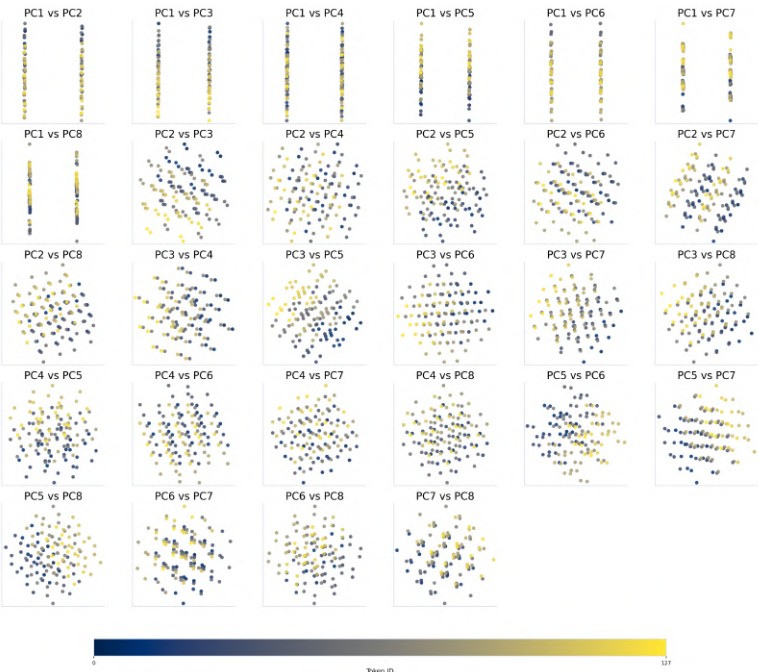

*Figure 39.* PCA projection of embeddings of model trained on the HYPERCUBE graph metric completion.

### D.4.5. 2D LATTICE

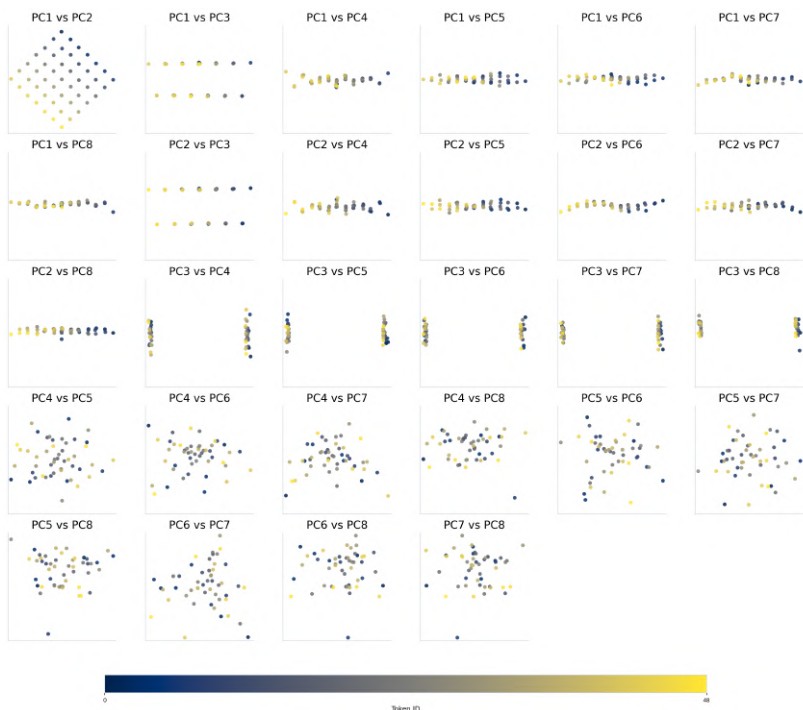

*Figure 40.* PCA projection of embeddings of model trained on the 2D LATTICE graph metric completion.

### D.4.6. 3D LATTICE

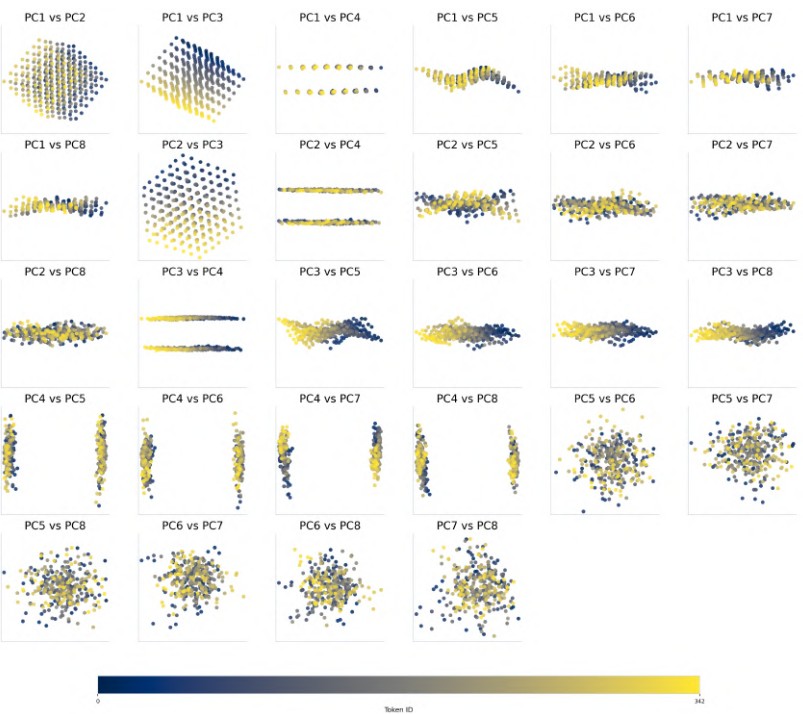

*Figure 41.* PCA projection of embeddings of model trained on the 3D LATTICE graph metric completion.

## D.5. Comparison

### D.5.1. 2D COMPARISON

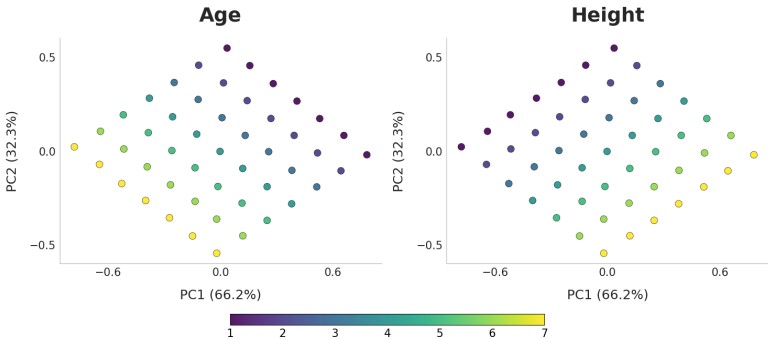

*Figure 42.* PCA projection of embeddings of model trained on the 2D COMPARISON.

### D.5.2. 3D COMPARISON

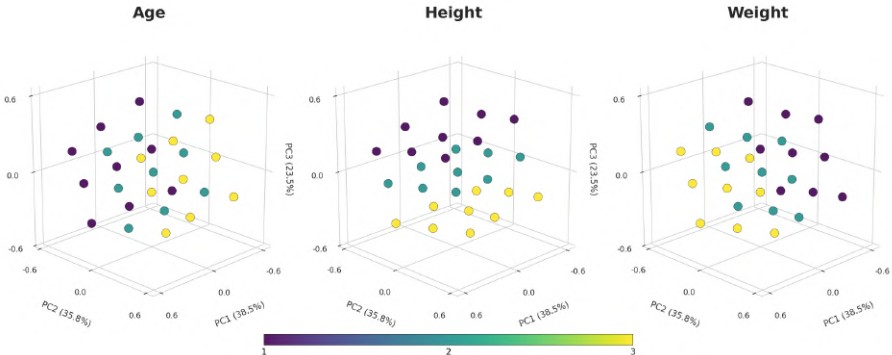

*Figure 43.* PCA projection of embeddings of model trained on the 3D COMPARISON.

# E. Accelerating Tasks using Symmetric Regularization

## E.1. Formulation

We first describe the formulation of each acceleration variant. Each variant augments the standard cross-entropy objective with an auxiliary loss $\mathcal{L}_{\text{aux}}$ that encourages task-specific structural consistency.

### E.1.1. GEOMETRY-PROMPTING

For geometry-prompting, we directly regularize the learned embedding space. Let $E \in \mathbb{R}^{V \times d}$ denote the embedding matrix, where $V$ is the number of entities, and let $E_i$ denote the embedding of entity $i$. We consider the following auxiliary objectives:

$$\mathcal{L}_{\text{nuc}} = \|E\|_*, \qquad \mathcal{L}_{\text{lip}} = \frac{1}{V} \sum_i \|E_i - E_{i+1}\|_2^2, \qquad \mathcal{L}_{\text{ent}} = -\sum_i \pi_i \log \pi_i.$$

Here, $\mathcal{L}_{\text{nuc}}$ encourages low-rank structure in the embedding space, $\mathcal{L}_{\text{lip}}$ promotes local smoothness between neighboring entity embeddings, and $\mathcal{L}_{\text{ent}}$ controls the entropy of the embedding-induced distribution $\pi$.

### E.1.2. SYMMETRY-PROMPTING

For symmetry-prompting, we use the same task-specific symmetry metrics described in Appendix A as auxiliary regularizers.

## E.2. Full Results

### E.2.1. MODULAR ARITHMETIC

Note that Consistency refers to joint enforcement of commutativity and associativity.

### E.2.2. ADDITION: $x + y$

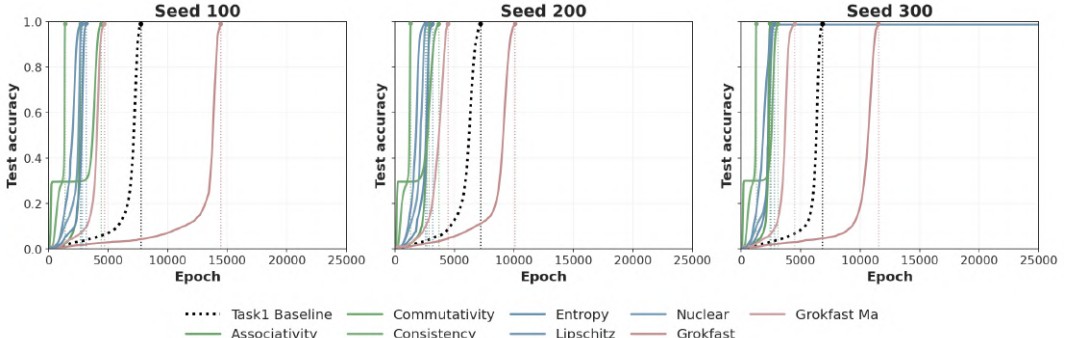

*Figure 44.* Convergence speed on modular addition when using various regularization.

### E.2.3. SUBTRACTION: $x - y$

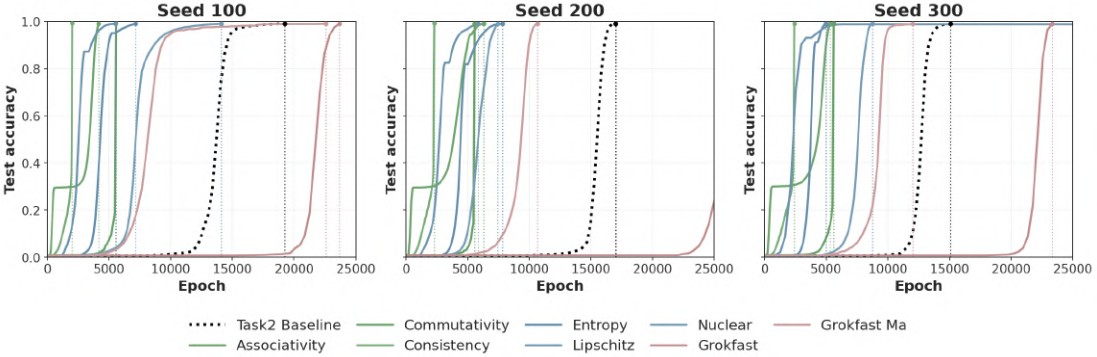

*Figure 45.* Convergence speed on modular subtraction when using various regularization.

### E.2.4. SQUARED SUM: $(x + y)^2$

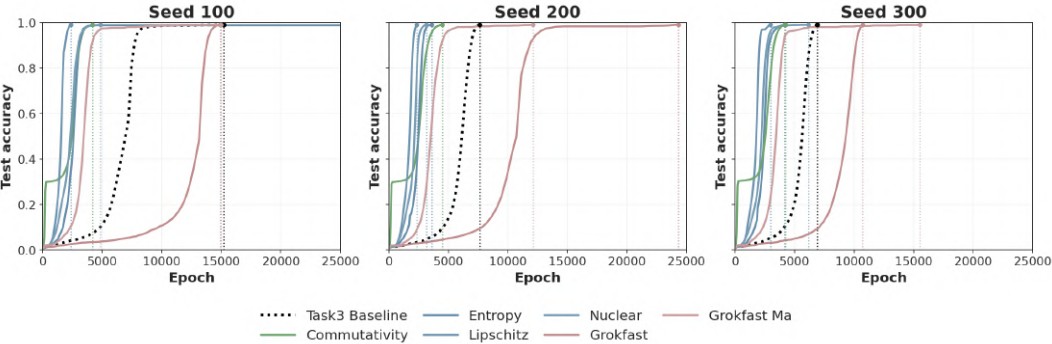

*Figure 46.* Convergence speed on modular squared sum when using square sum.

### E.2.5. CUBED SUM: $(x + y)^3$

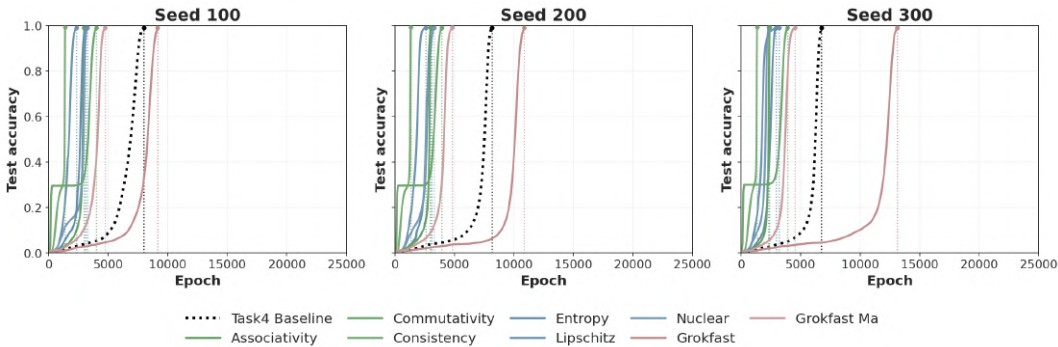

*Figure 47.* Convergence speed on modular cubed sum when using various regularization.

### E.2.6. MULTIPLICATION: $x \cdot y$

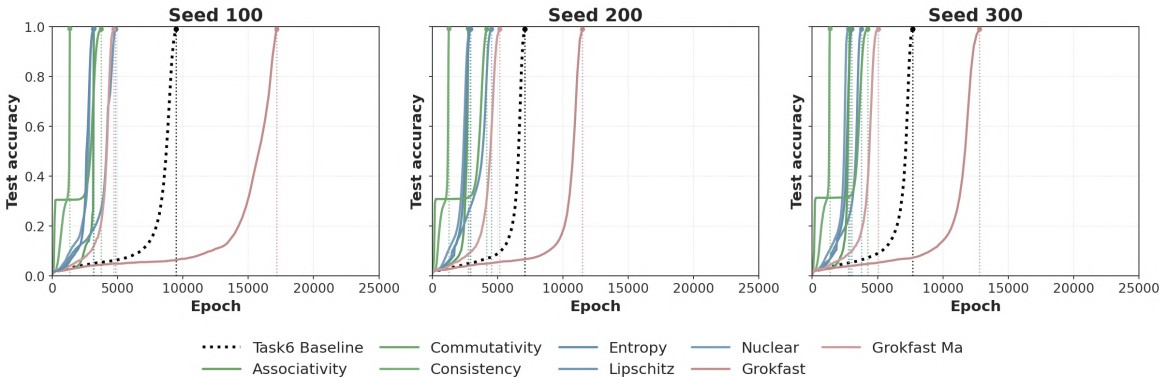

*Figure 48.* Convergence speed on modular multiplication when using various regularization.

### E.2.7. AFFINE–BILINEAR: $x + xy + y$

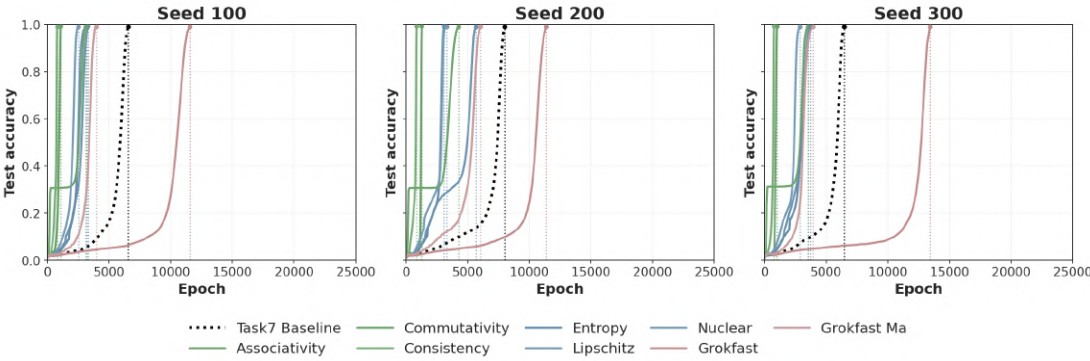

*Figure 49.* Convergence speed on modular affine–bilinear when using various regularization.

## E.3. graph metric completion

### E.3.1. CYCLE

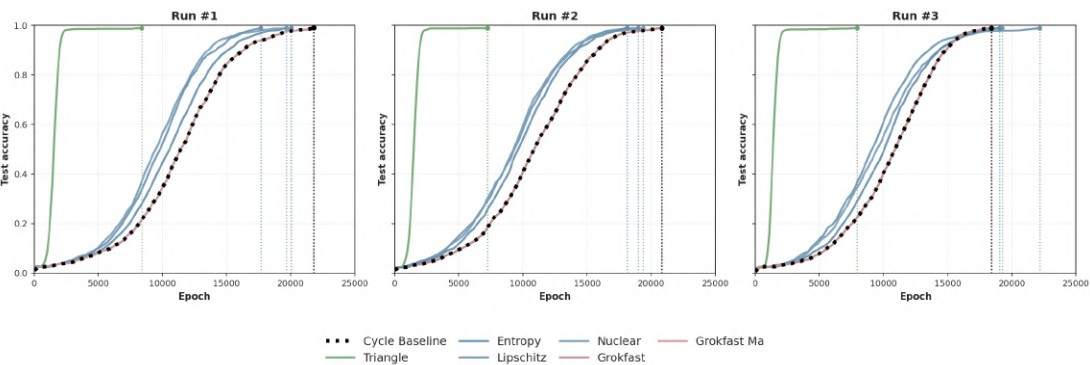

*Figure 50.* Convergence speed on **cycle** graph metric completion when using various regularization.

### E.3.2. PATH

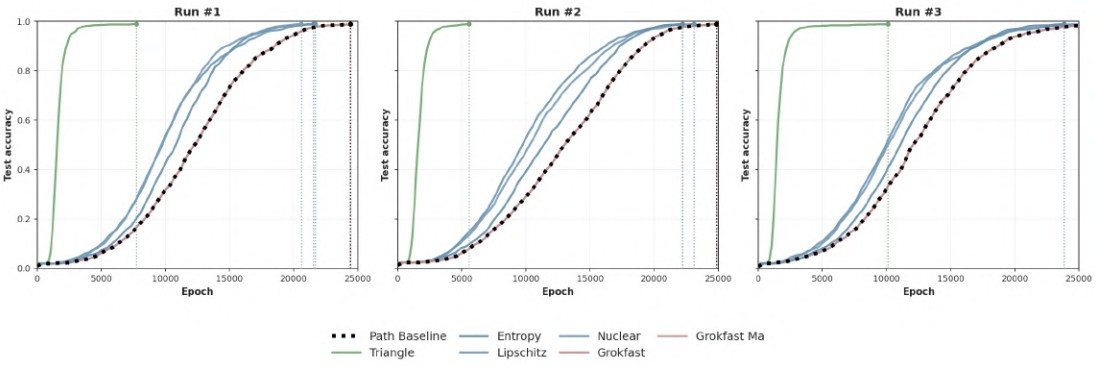

*Figure 51.* Convergence speed on **path** graph metric completion when using various regularization.

### E.3.3. CYLINDER

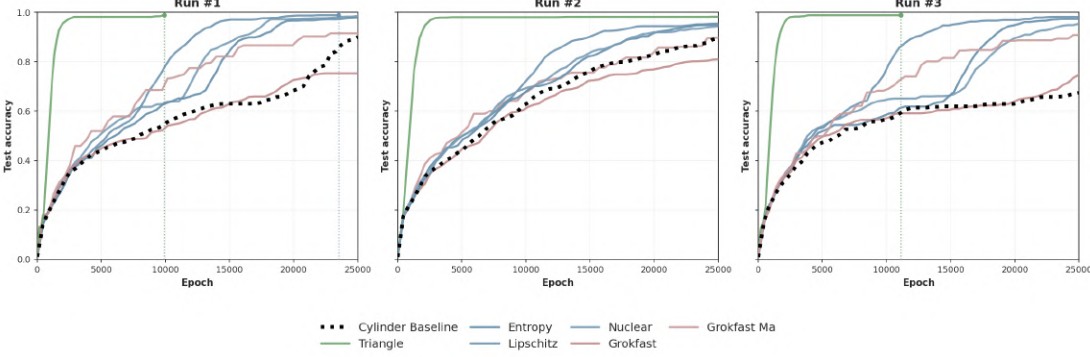

*Figure 52.* Convergence speed on **cylinder** graph metric completion when using various regularization.

### E.3.4. HYPERCUBE

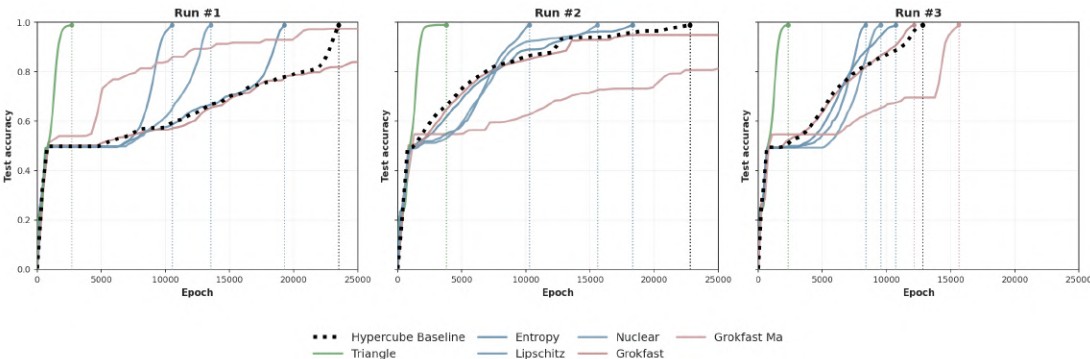

*Figure 53.* Convergence speed on **hypercube** graph metric completion when using various regularization.

### E.3.5. 2D LATTICE

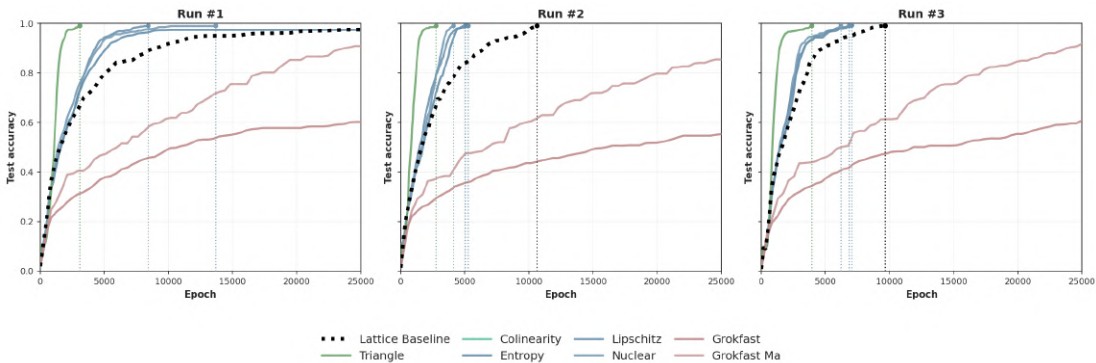

*Figure 54.* Convergence speed on **2d lattice** graph metric completion when using various regularization.

### E.3.6. 3D LATTICE

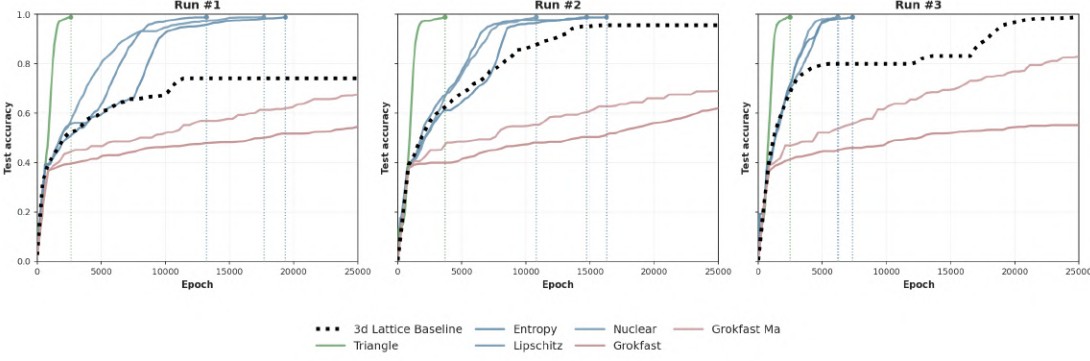

*Figure 55.* Convergence speed on **3d lattice** graph metric completion when using various regularization.

## E.4. Comparison

### E.4.1. 2D COMPARISON

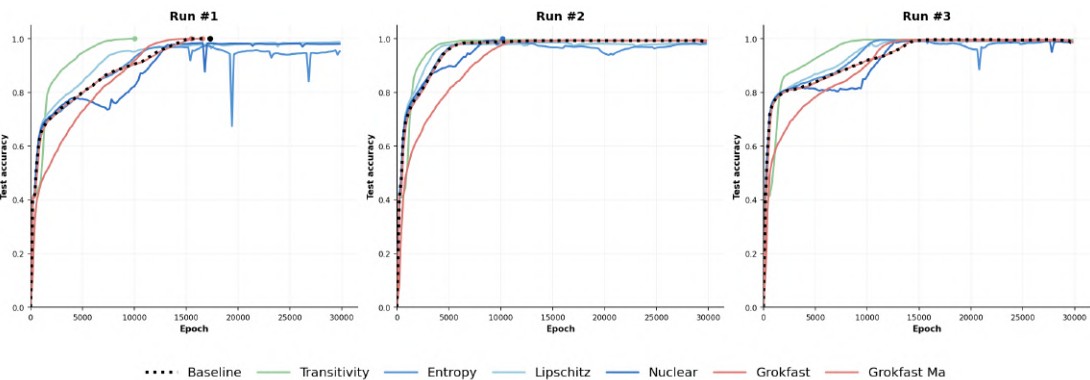

*Figure 56.* Convergence speed on **Comparison (2D)** when using various regularization.

### E.4.2. 3D COMPARISON

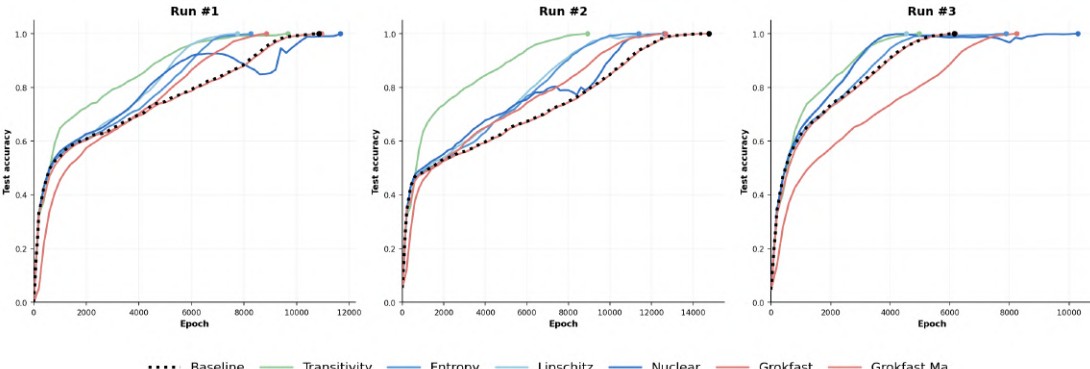

*Figure 57.* Convergence speed on **Comparison (3D)** when using various regularization.

## F. Symmetry Suppression

To further assess the causal role of symmetry in generalization, we examine whether explicitly suppressing symmetry acquisition degrades performance or prevents convergence, similar to (Han et al., 2026). Specifically, we compare the vanilla model against models trained with positive and negative weights on the task-specific symmetry constraint. As shown in Figure 58, encouraging the relevant symmetry accelerates convergence, whereas suppressing it generally prevents the model from reaching perfect generalization. This supports the view that symmetry acquisition is not merely correlated with generalization, but plays an active role in driving it.

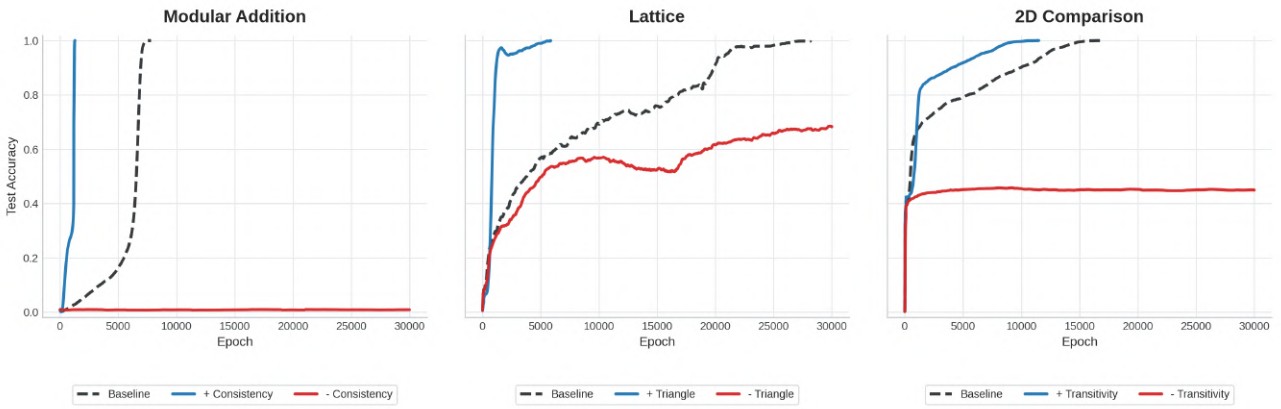

*Figure 58.* Convergence under symmetry promotion and suppression.

## G. Training Configuration

**Model Configuration.** We employ a standard single-layer ($L = 1$) decoder-only Transformer with $H = 4$ attention heads. The architecture uses an embedding dimension of $d_{\text{model}} = 128$, a feed-forward (MLP) dimension of $d_{\text{mlp}} = 512$, and a head dimension of $d_{\text{head}} = 32$. All models utilize ReLU activations and omit Layer Normalization.

**Training** All models are trained for a maximum of $100,000$ epochs using the AdamW optimizer.

For each task, sepcifically, we employ following configurations:

**Modular Arithmetic.** Following standard protocols, we use modulus $p = 113$, a training split of $r_{\text{train}} = 0.3$, and a weight decay of $w = 1$.

**Graph Metric Completion.** We increase the training split to $r_{\text{train}} = 0.8$ and weight decay to $w = 3$. We evaluate performance across six distinct graph structures:

- (i) 2D Grid Lattices ($7 \times 7$)   (ii) 3D Grid Lattices ($7 \times 7 \times 7$)   (iii) Simple Cycles ($N = 64$)
- (iv) Path Graphs ($N = 64$)   (v) Cylinders ($7 \times 7$ periodic)   (vi) 7D Hypercubes ($2^7$ nodes)

**Comparison.** we use a reduced training split of $r_{\text{train}} = 0.2$ with weight decay $w = 1$. The dataset consists of entities arranged in either 2D ($7 \times 7$) or 3D ($3 \times 3 \times 3$) structures. Each entity is assigned a unique combination of attributes (height, weight, age), sampled such that values lie within a bounded range $[v_{\text{min}}, v_{\text{max}}]$.

**Acceleration.** For the acceleration experiments, we use the same setting as vanilla run, including the weight decay setting above used for each task. For 3D Lattice, we used $4 \times 4 \times 4$ setting to reduce computational burden.

# H. Towards Automatic Discovery of Symmetries

A limitation of the symmetry-prompting objectives used in the main experiments is that they assume access to candidate task symmetries. While this is useful for isolating the role of symmetry in controlled algorithmic settings, the relevant symmetry structure may not be known a priori in more realistic settings. Here, we provide an initial step toward automatically identifying such symmetries.

**Formulation.** The simplest class of task symmetries can be described as label-preserving transformations of the input space. Let $\mathcal{X}$ be the input space and let $f : \mathcal{X} \to \mathcal{Y}$ be the target function. A transformation $T : \mathcal{X} \to \mathcal{X}$ is an input-level symmetry if

$$f(Tx) = f(x) \qquad \text{for all } x \in \mathcal{X}.$$

For example, commutativity can be written as $T(a, b) = (b, a)$. Given a trained model $p_\theta(\cdot \mid x)$, we measure how strongly the model respects a candidate transformation $T$ by

$$\mathcal{D}_{\text{sym}}(T) = \mathbb{E}_{x \sim \mathcal{D}} \left[ D_{\text{SKL}} \big( p_\theta(\cdot \mid x), p_\theta(\cdot \mid Tx) \big) \right], \qquad D_{\text{SKL}}(p, q) = \frac{1}{2} D_{\text{KL}}(p \| q) + \frac{1}{2} D_{\text{KL}}(q \| p).$$

Transformations with low $\mathcal{D}_{\text{sym}}(T)$ are candidate symmetries that the model treats as approximately label-preserving.

**Procedure.** This gives a simple procedure for symmetry discovery: define a search space of transformations or computation paths, evaluate each candidate using the corresponding discrepancy measure, and treat low-discrepancy candidates as potential symmetries. Such candidates should be viewed as model-discovered symmetry hypotheses rather than proof of true task symmetries, and should be further validated using held-out data or explicit task labels.

**Example.** As a proof of concept, we consider a counting task where each input is $[c_1, c_2, c_3, q]$. The context tokens $c_1, c_2, c_3$ are drawn from $\{1, \ldots, 10\}$, and $q$ is the query token. The target output is the number of times $q$ appears among $c_1, c_2, c_3$. We train the model on $50\%$ of all input–output pairs.

Since the label depends only on the multiset of context tokens, the task is invariant to permutations of $(c_1, c_2, c_3)$ while keeping $q$ fixed. Thus, for length-three contexts, the relevant input-level symmetry is the permutation group $S_3$. We evaluate candidate permutations using the symmetric KL discrepancy between the model outputs on the original and transformed inputs. Permutations in $S_3$ yield much lower discrepancy than transformations that move the query token, indicating that the procedure recovers the expected positional invariance. However, this example only covers simple input-level invariance and discovering more complex, higher-order symmetries remains an important direction for future work.

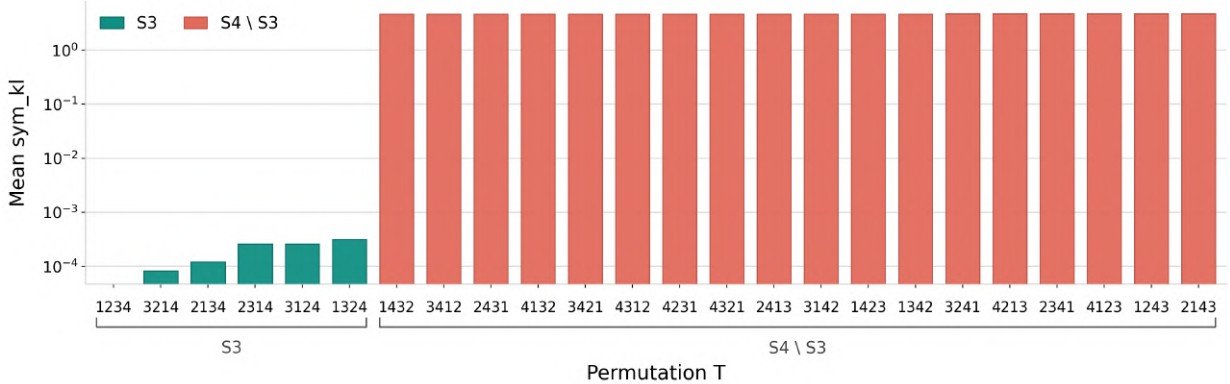

*Figure 59.* Permutations over context tokens ($S_3$) yield near-zero discrepancy, while permutations that move the query token ($S_4 \setminus S_3$) yield much larger discrepancy.

*Table 3.* Predicted embedding geometry and associated algorithm of various modular arithmetic tasks. Note that $q \in \mathbb{N}$, $\gcd(p, q) = 1$ and $g$ is the generator of multiplicative group $\mathbb{Z}_p^\times$, $\mathrm{ind}_g(\cdot)$ denote the discrete logarithm with base $g$, and $\log(\cdot)$ denote the complex logarithm.

| Task | Embedding function and associated algorithm | Embedding geometry | Notes |
|---|---|---|---|
| $x + y = z \pmod{p}$ | $\phi(x) = e^{2\pi i q x/p}$ 
 $\psi(x) = p \log(x)/(2\pi i q)$ 
 $z = \psi(\phi(x)\phi(y))$ | Circle (closed helix) 
 $\angle(i, i+1) = \frac{2\pi q}{p}$ | $\phi(x)\phi(y) = \phi(z)$ |
| $x - y = z \pmod{p}$ | $\phi(x) = e^{2\pi i q x/p}$ 
 $\psi(x) = p \log(x)/(2\pi i q)$ 
 $z = \phi(x)\phi(y)^{-1}$ | Circle (closed helix) 
 $\angle(i, i+1) = \frac{2\pi q}{p}$ | $\phi(x)\phi(y)^{-1} = \phi(z)$ |
| $(x+y)^\alpha = z \pmod{p} \; \alpha \in \mathbb{N}$ | $\phi(x) = e^{2\pi i q x/p}$ 
 $\psi(x) = h_\alpha(p \log(x)/(2\pi i q))$ 
 $z = \phi(x)\phi(y)$ | Circle (closed helix) 
 $\angle(i, i+1) = \frac{2\pi q}{p}$ | $h_\alpha : \mathbb{Z}_p^\times \to \mathbb{Z}_p^\times,$ 
 $h_\alpha(x) : x \mapsto x^\alpha \pmod{p}$ 
 $\phi(x)\phi(y) = \phi(z)$ |
| $x \times y = z \pmod{p}$ | $\phi(x) = e^{2\pi i q \cdot \mathrm{ind}_g(x)/p}$ 
 $\psi(x) = g^{p \cdot \log(x)/(2\pi i q)}$ 
 $z = \psi(\phi(x)\phi(y))$ | Circle (closed helix) 
 $\angle(g^i, g^{i+1}) = \frac{2\pi q}{p}$ | $\phi(x)\phi(y) = \phi(z)$ |
| $x + xy + y = z \pmod{p}$ | $\phi(x) = e^{2\pi i q \cdot \mathrm{ind}_g(x+1)/p}$ 
 $\psi(x) = g^{p \cdot \log(x)/(2\pi i q)}$ 
 $z = \psi(\phi(x)\phi(y)) - 1$ | Circle (closed helix) 
 $\angle(g^i - 1, g^{i+1} - 1) = \frac{2\pi q}{p}$ | $\phi(x)\phi(y) = \phi(z)$ |

# I. Symmetries in modular arithmetic

We present the predicted embedding geometry of various modular arithmetic tasks. For modular addition, the exact algorithm using trigonometric identities has been reported in (Nanda et al., 2023; Zhong et al., 2023; Park et al., 2025). If we consider the embedding function $\phi : \mathbb{Z}_p \to \mathbb{C}$ and outer function $\psi : \mathbb{C} \to \mathbb{Z}_p$, there is a simple identity to formulate modular addition.

$$\phi(x) = e^{2\pi i q x/p},$$

$$\psi(x) = \frac{p}{2\pi i q} \log(x), \qquad q \in \mathbb{Z}_p, \; \gcd(p, q) = 1,$$

$$z = \psi(\phi(x)\phi(y)),$$

$$z = \psi\left(e^{2\pi i q(x+y)/p}\right) = x + y \pmod{p}.$$

where $\log(\cdot)$ denotes complex logarithm. The embedding $\phi(\cdot)$ says that the embedding geometry $\{\phi(x) : x \in \mathbb{Z}_p\}$ have shape of the circle with the consistent angle between adjacent tokens being $\frac{2\pi q}{p}$.

For modular multiplication, $\mathbb{Z}_p^\times = \mathbb{Z}_p \setminus \{0\}$ become cyclic group with some generator $g \in \mathbb{Z}_p^\times$ (*i.e.* $\mathbb{Z}_p^\times = \{g^i : i = 0, 1, ..., p-1\}$ ). Hence we have embedding function $\phi : \mathbb{Z}_p^\times \to \mathbb{C}$ and outer function $\psi : \mathbb{C} \to \mathbb{Z}_p^*$, there is a simple identity to formulate modular addition.

$$\phi(x) = e^{2\pi i q \, \mathrm{ind}_g(x)/p},$$

$$\psi(x) = \frac{p}{2\pi i q} \log(x), \qquad q \in \mathbb{Z}_p, \; \gcd(p, q) = 1,$$

$$z = \psi(\phi(x)\phi(y)),$$

$$z = \psi\left(e^{2\pi i q(\mathrm{ind}_g(x)+\mathrm{ind}_g(y))/p}\right) = x \times y \pmod{p}.$$

where $\mathrm{ind}_g(\cdot)$ denotes the discrete logarithm with base $g$ in $\mathbb{Z}_p$. For $T_6 : z = x + xy + y \pmod{p}$, we can rewrite it as $z + 1 = (x + 1)(y + 1) \pmod{p}$. Hence the embedding is similar to multiplication with translation by 1.

Similarly, we present the predicted embedding geometry and algorithm of other various modular arithmetic tasks which are presented in Table 3 in Appendix I.

We present the embedding geometry and proxy symmetries of various modular arithmetic in Tables 3 and 1.

## J. Theoretical observation

### J.1. Theoretical observation on modular arithmetic

First, we state the Pontryagin duality theorem in Lie groups.

**Definition J.1** (Pontryagin dual). For a locally compact abelian group $G$, the Pontryagin dual is the group $\hat{G}$ of continuous group homomorphisms from $G$ to the circle group $\mathbb{T}$, *i.e.*

$$\hat{G} := \mathrm{Hom}(G, \mathbb{T}).$$

**Lemma J.2.** *The Pontryagin dual of* $\mathbb{T}$ *is* $\mathbb{Z}$ *via the map*

$$\Phi : \mathbb{Z} \cong \hat{\mathbb{T}}$$
$$n \mapsto \left(z \mapsto e^{2\pi i n z}\right).$$

**Lemma J.3.** *The Pontryagin dual of* $\mathbb{Z}$ *is* $\mathbb{T}$ *via the map*

$$\Phi : \mathbb{T} \cong \hat{\mathbb{Z}}$$
$$e^{i\theta} \mapsto \left(n \mapsto e^{in\theta}\right).$$

**Theorem J.4** (Pontryagin duality (Pontrjagin, 1934; Morris, 1977; Hewitt & Ross, 2013)). *Let $G$ be a locally compact abelian topological group. Then there is a canonical isomorphism such that*

$$G \cong \hat{\hat{G}},$$

*where $\hat{\hat{G}}$ is Pontryagin double dual of $G$.*

If topological abelian group $G$ is one-dimensional, $G$ is isomorphic to the closed helix.

### J.2. Proof of Proposition 7.1

**Proposition.** Let $\mathcal{M}$ be an one-dimensional compact abelian topological group and $\mathbb{Z}_p$ is continuously embedded in $\mathcal{M}$. Then $\mathcal{M}$ is isomorphic to $S^1$ embedded in high-dimensional torus $\mathbb{T}^D$ (closed helix)

$$\Phi : \mathcal{M} \cong S^1 \hookrightarrow \mathbb{T}^D.$$

Hence $\Phi(\mathbb{Z}_p) \hookrightarrow \Phi(\mathcal{M})$ is also confined in the closed helix.

*Proof.* Note that the one-dimensional only connected, compact, abelian topological group is the circle group $\mathbb{T} \cong \mathbb{R}/\mathbb{Z} \cong [0,1]/\{0 \sim 1\}$. By Theorem J.4, we have

$$\Phi : \mathbb{T} \cong \hat{\hat{\mathbb{T}}} \cong \hat{\mathbb{Z}}$$

via the map

$$\Phi(x) = \left(n \mapsto e^{2\pi i n x}\right), \quad n \in \mathbb{Z}.$$

Therefore,

$$\Phi(\mathbb{T}) = \prod_{n \in \mathbb{Z}} n\mathbb{T} \subset \mathbb{T}^\infty,$$

where $n$-th each component denotes the rotated by $n$ times in infinite-dimensional torus. We note that $\Phi(\mathbb{T})$ can be embedded into some high-dimensional torus $\mathbb{T}^D$

$$\Phi(\mathbb{T}) \cong (n_1\mathbb{T}, n_2\mathbb{T}, \ldots, n_D\mathbb{T}) \subset \mathbb{T}^D,$$

if $\gcd(n_1, \ldots, n_D) = 1$. Therefore, $\mathcal{M}$ is isomorphic to $S^1 \hookrightarrow \mathbb{T}^D$ in some high-dimensional torus $\mathbb{T}^D$.

$\square$

### J.3. Theoretical observation on graph metric completion and comparison tasks

In this section, we describe the embedding geometry in graph completion and comparisons task.

J.3.1. PROOF OF PROPOSITION 7.2

**Proposition.** Let $\mathcal{X} = \{v_1, \ldots, v_n\}$ be a one-dimensional Path or Cycle graph (see Figure 3) with $n$ vertices ($n \geq 3$) and $\mathcal{Z} \in \{\mathbb{R}, \mathbb{T}\}$ be a embedding space. Let $\Phi : G \to \mathcal{Z}$ be embedding with partial distance information

$$d(\Phi(v_i), \Phi(v_{i+1})) = 1, \; i = 1, \ldots n-1,$$
$$d(\Phi(v_1), \Phi(v_n)) = 1, \; \text{if } \mathcal{X} \text{ is Cycle.}$$

If $\Phi(\cdot)$ preserve the triangular symmetry, then $\Phi(\mathcal{X})$ has its graph structure in embedding space $\mathbb{R}$.

*Proof.* **Case 1:** $\mathcal{X}$ is Path.

**Case 1-1:** $\mathcal{Z} = \mathbb{R}$.
First suppose $n = 3$ and $\mathcal{X} = \mathbb{R}$ with the standard metric. Let $v_1, v_2, v_3$ be the vertices of $\mathcal{X}$. Consider the embedding $\Phi : \mathcal{X} \to \mathbb{R}$. Without loss of generality, let $\Phi(v_1) = 0$. Since $\Phi(\cdot)$ preserves the triangular symmetry, we have $d(v_1, v_3) = d(v_1, v_2) + d(v_2, v_3) = 2$. Then $\Phi(v_2) = \pm 1$ because $d(\Phi(v_1), \Phi(v_2)) = 1$. Without loss of generality, let $\Phi(v_1) = 1$. Then we have $\Phi(v_3) = 2$ because $d(\Phi(v_2), \Phi(v_3)) = 1, d(\Phi(v_1), \Phi(v_3)) = 2$. Therefore, $v_1, v_2, v_3$ should be aligned in a order.
For general $n \geq 4$, by applying the mathematical induction, we achieve the desired result.

**Case 1-2:** $\mathcal{Z} = \mathbb{T}$.
Let $\mathbb{T} = \{e^{2\pi i \theta / K} : \theta \in \mathbb{R}\}$ with metric $d(e^{i\theta_1}, e^{i\theta_2}) = \min_{m \in \mathbb{Z}} |(\theta_1 - \theta_2) - Km|$ for some large $K \gg n$. First suppose $n = 3$. Let $v_1, v_2, v_3$ be the vertices of $\mathcal{X}$. Consider the embedding $\Phi : G \to \mathbb{T}$, and Let $\Phi(v_i) = e^{2\pi i \theta_i / K} \in \mathbb{T}$. Without loss of generality, let $\theta_1 = 0$. Since $\Phi(\cdot)$ preserves the triangular symmetry, we have $d(v_1, v_3) = d(v_1, v_2) + d(v_2, v_3) = 2$.

Then $\theta_2 = \pm 1$ because $d(\Phi(v_1), \Phi(v_2)) = 1$. Without loss of generality, let $\theta_2 = 1$. Then we have $\theta_3 = 2$ because $d(\Phi(v_2), \Phi(v_3)) = 1, d(\Phi(v_1), \Phi(v_3)) = 2$. Therefore, $v_1, v_2, v_3$ should be aligned in a order. For general $n \geq 4$, by applying the mathematical induction, we achieve the desired result. We also note that if $\mathbb{T}$ has different metric like pullback metric as

$$d_k(e^{i\theta_1}, e^{i\theta_2}) = \min_{m \in \mathbb{Z}} |k(\theta_1 - \theta_2) - Km|, \quad \frac{k}{K} \in \mathbb{Z}$$

for some $k \in \mathbb{R}$, the we have different embeddings.

**Case 2:** $\mathcal{X}$ is Cycle.
In this case, $\mathcal{Z} = \mathbb{R}$ is impossible. If $\mathcal{Z} = \mathbb{T}$, let $\mathbb{T} = \{e^{2\pi i \theta / K} : \theta \in \mathbb{R}\}$ with metric $d(e^{i\theta_1}, e^{i\theta_2}) = \min_{m \in \mathbb{Z}} |(\theta_1 - \theta_2) - Km|$ for some large $K \gg n$. In Cycle graph, note that $d(v_1, v_3) = 1$. Let $\Phi(v_i) = e^{2\pi i \theta_i / K} \in \mathbb{T}$. Without loss of generality, assume $\theta_1 = 0$ and $\theta_2 = 1$. Then we have $\theta_2 = -1$ from $d(v_1, v_3) = 1$ and $\theta_2 = 2$ from $d(v_2, v_3) = 1$. Hence $e^{2\pi i (-1)/K} = e^{2\pi i (2)/K}$ leading to $e^{2\pi i (3)/K} = 1$, $K \equiv 0 \pmod 3$. Hence $\Phi(v_1), \Phi(v_2), \Phi(v_3)$ has cyclic shape in $\mathbb{T}$. For general $n \geq 4$, by applying the mathematical induction, we achieve the desired result.

We also note that if $\mathbb{T}$ has different metric like pullback metric as

$$d_k(e^{i\theta_1}, e^{i\theta_2}) = \min_{m \in \mathbb{Z}} |k(\theta_1 - \theta_2) - Km|, \quad \frac{k}{K} \in \mathbb{Z}$$

for some $k \in \mathbb{R}$, the we have different embeddings.

$\square$

### J.3.2. PROOF OF PROPOSITION 7.3

**Proposition.** Let $\mathcal{X} = \{v_1, \ldots, v_n\}$ be a set of nodes in 1D comparison task (see Section 3.3) with $n \geq 3$ nodes. Let $\Phi : \mathcal{X} \to \mathbb{R}$ be embedding with partial relation information

$$\Phi(v_i) < \Phi(v_{i+1}), \ i = 1, \ldots n - 1.$$

If $\Phi(\cdot)$ preserve the transitivity, then $\Phi(\mathcal{X})$ has a grid structure in embedding space $\mathbb{R}$.

*Proof.* Suppose $n = 3$. Let $v_1, v_2, v_3$ be nodes of $\mathcal{X}$ with $v_1 < v_2 < v_3$. Consider the embedding $\Phi : \mathcal{X} \to \mathbb{R}$. Without loss of generality, let $\Phi(v_1) = 0$. Since $\Phi(\cdot)$ preserves the transitivity, we have $\Phi(v_1) < \Phi(v_3)$ from $\Phi(v_1) < \Phi(v_2)$ and $\Phi(v_2) < \Phi(v_3)$. Hence we have $\Phi(v_1) < \Phi(v_2) < \Phi(v_3)$. For general $n \geq 4$, by applying the mathematical induction, we achieve the desired result. $\square$

### J.4. The proof of Proposition 7.4

**Proposition.** Let $G = \{0, 1, \ldots, p - 1\}$ and let $\odot$ be a binary operation on $G$. Assume that $(G, \odot)$ is a cyclic group of order $p$, with identity element 0 and generator 1. Then there exists a subset $M \subset G \times G$, with $|M| = 2p - 1$, such that the values of $\odot$ on $M$, namely $\tilde{M} := \{(a, b, c) \in G^3 : (a, b) \in M, \ a \odot b = c\}$, uniquely determine the full Cayley table of $(G, \odot)$. Consequently, the fraction of entries required to reconstruct the operation is

$$\frac{|M|}{|G \times G|} = \frac{2p - 1}{p^2} \sim \frac{2}{p}.$$

*Proof.* Define

$$M := \{(a, 0) : a \in G\} \ \cup \ \{(a, 1) : a \in G\}.$$

Then $|M| = p + (p - 1) = 2p - 1$.

From $\tilde{M}$, we know

$$a \odot 0 = a \qquad \text{for all } a \in G,$$

and

$$a \odot 1 = a^+ \qquad \text{for all } a \in G, \ a \neq 0,$$

where $a^+ := a \odot 1$. Moreover, since $1 \odot 0 = 1$ is known from $\tilde{M}$, commutativity gives

$$0 \odot 1 = 1.$$

Hence the value $a \odot 1$ is known for every $a \in G$. In other words, the map

$$a \mapsto a^+ := a \odot 1$$

is completely determined by $\tilde{M}$.

In particular, the sequence

$$0, \quad 1, \quad 1^+, \quad 1^{++}, \quad \ldots$$

is determined by $\tilde{M}$. Since 1 is a generator of $G$, this sequence exhausts all elements of $G$.

Let $\mathcal{D} \subset G \times G$ denote the subset uniquely determined by $\tilde{M}$. By commutativity,

$$\mathcal{D}_1 := \{(a, b) : a \in G, \ b \in \{0, 1\}\} \cup \{(b, a) : a \in G, \ b \in \{0, 1\}\}$$

is uniquely determined by $\tilde{M}$, and we have

$$\mathcal{D}_1 \subset \mathcal{D}.$$

Since $(1, 1) \in \mathcal{D}_1$, we know
$$1 \odot 1 = 1^+.$$

Since $(a, 1) \in \mathcal{D}_1$, we know
$$a \odot 1 = a^+.$$

Hence, by associativity,
$$a \odot 1^+ = a \odot (1 \odot 1) = (a \odot 1) \odot 1 = a^+ \odot 1.$$

Since $(a^+, 1) \in \mathcal{D}_1$, we know
$$a^+ \odot 1 = a^{++},$$

where $a^{++} := a^+ \odot 1$. Hence
$$a \odot 1^+ = a^{++}$$

is uniquely determined, and therefore $(a, 1^+) \in \mathcal{D}$. By commutativity, $(1^+, a) \in \mathcal{D}$. Therefore,

$$\mathcal{D}_2 := \{(a, b) : a \in G, \ b \in \{0, 1, 1^+\}\} \cup \{(b, a) : a \in G, \ b \in \{0, 1, 1^+\}\} \subset \mathcal{D}.$$

By a similar argument, let
$$1^{++} := 1^+ \odot 1.$$

Since $(1^+, 1) \in \mathcal{D}_2$, we know
$$1^+ \odot 1 = 1^{++}.$$

Hence, by associativity,
$$a \odot 1^{++} = a \odot (1^+ \odot 1) = (a \odot 1^+) \odot 1 = a^{++} \odot 1.$$

Since $(a^{++}, 1) \in \mathcal{D}_2$, we know
$$a^{++} \odot 1 = a^{+++},$$

where $a^{+++} := a^{++} \odot 1$. Hence
$$a \odot 1^{++} = a^{+++}$$

is uniquely determined, and therefore $(a, 1^{++}) \in \mathcal{D}$. By commutativity, $(1^{++}, a) \in \mathcal{D}$. Therefore,

$$\mathcal{D}_3 := \{(a, b) : a \in G, \ b \in \{0, 1, 1^+, 1^{++}\}\} \cup \{(b, a) : a \in G, \ b \in \{0, 1, 1^+, 1^{++}\}\} \subset \mathcal{D}.$$

Inductively, for every element $b$ in the sequence

$$0, \quad 1, \quad 1^+, \quad 1^{++}, \quad \ldots,$$

the values $a \odot b$ and $b \odot a$ are uniquely determined for all $a \in G$. Since this sequence exhausts $G$, we obtain

$$G \times G \subset \mathcal{D}.$$

Since $\mathcal{D} \subset G \times G$ by definition, we conclude that

$$\mathcal{D} = G \times G.$$

This proves that $\tilde{M}$ uniquely determines the full Cayley table of $(G, \odot)$. $\qquad\square$

