# OpenReview forum: "Intrinsic Task Symmetry Drives Generalization in Algorithmic Tasks"
_ICML.cc/2026/Conference — ICML 2026 regular_

### Official Review · Reviewer_rWvz · 2026-03-11

**Soundness:** 2
**Presentation:** 4
**Significance:** 4
**Originality:** 2
**Overall Recommendation:** 3
**Confidence:** 4

**Summary:**

The authors study grokking through the lens of symmetries in task structure. They train several models on different tasks (metric completion, comparison, modular arithmetic) with different symmetries. Through a combination of dimensionality reduction and observations of the test accuracy over training, the authors suggest that learning can generally be decomposed into: (1) learning the training data, (2) learning the symmetries, and (3) learning to use those symmetries. These insights are supported by extensive numerical experiments including: using dimensionality reduction, adding extra terms in the loss to promote symmetries, as well as some theoretical results regarding graphs, algebras and topologies.

**Compliance With Llm Reviewing Policy:**

Affirmed.

**Final Justification:**

The authors have answered my questions. Yet, the theoretical contributions are still, in my opinion, limited despite taking a significant portion of the paper. I have therefore kept my original score.

**Key Questions For Authors:**

### Questions / Comments

**1. Clarification of theoretical contribution.** To my understanding, the theoretical results regarding modular arithmetic and graph completion tasks (Propositions 7.1–7.4) are standard results from topology, graph theory, and arithmetic. Could the authors confirm this and perhaps provide more systematic citations to relevant textbooks? More generally, could the authors clarify what their theoretical contribution is, if any?

**2. Discussion of possible classes of symmetries.** The main conceptual step of the paper is to generalize existing analyses of grokking to new tasks with different types of symmetries. However, the analyses obviously do not cover all symmetries. They chose those coming from graph structure and relations. Could the authors comment on how one might approach a more systematic exploration of different classes of symmetries?

**3. Link to the study of optimization dynamics.** Could the authors discuss the link to more general theories of *optimization dynamics* and their model under different types of symmetries, akin to what is done in [1-2].

**4. Link to manifolds and Lie groups.** How does the continuity (or lack thereof) of input data play into this framework? For example, cyclic operations can be represented either as actions over finite groups or as operations over continuous manifolds. Do the authors expect that the model might learn different representations, and possibly exhibit different grokking behavior, in these two cases? Other works have suggested generalization on continuous manifolds due to symmetries in their Riemannian geometry, including in arithmetic tasks [3-4].

**5. Causal experiments.** The authors provide a more direct experiment where they add an extra term in the loss to  promote symmetries. The authors claim this to be a causal link; however I believe that a better experiment to probe for such a causal effect would be the opposite one: namely penalize negatively (or prevent altogether) symmetries; akin to what is done in [5] with the neural collapse term.

*The paper is nevertheless interesting, and if the authors address the questions above I will increase my score.*

### References

[1] Kumar, T., Bordelon, B., Gershman, S. J., & Pehlevan, C. (2023, September). Grokking as the transition from lazy to rich training dynamics. In The twelfth international conference on learning representations.

[2] Mallinar, N., Beaglehole, D., Zhu, L., Radhakrishnan, A., Pandit, P., & Belkin, M. (2024). Emergence in non-neural models: grokking modular arithmetic via average gradient outer product. arXiv preprint arXiv:2407.20199.

[3] Zavatone-Veth, J. A., Yang, S., Rubinfien, J. A., & Pehlevan, C. (2023). How does training shape the Riemannian geometry of neural network representations?. arXiv preprint arXiv:2301.11375.

[4]  Brandon, J., Chadwick, A., & Pellegrino, A. (2025). Emergent Riemannian geometry over learning discrete computations on continuous manifolds. arXiv preprint arXiv:2512.00196.

[5] Han, T., Adilova, L., Petzka, H., Kleesiek, J., & Kamp, M. (2025). Flatness is necessary, neural collapse is not: Rethinking generalization via grokking. arXiv preprint arXiv:2509.17738.

**Limitations:**

yes

**Strengths And Weaknesses:**

**Soundness.** The paper claims to address *why* geometric representations emerge during learning and *how* these representations are linked to improved generalization performance late in training. I find the general hypothesis that learning can be decomposed into multiple stages plausible. However, **the numerical results do not convincingly explain either why or how these stages unfold during learning.** In particular, the authors do not provide theoretical results regarding the learning dynamics of their models, nor do they directly probe specific mechanisms within the model (akin to the ones explained in [1-2]). In other words, the results are mostly phenomenological and rely primarily on dimensionality reduction methods. This is the main reason why I do not provide a higher soundness score.

**Presentation.** The paper is overall satisfactorily presented. The authors systematically provide visual examples of the concepts they explain, and the writing is clear. The organization of the sections could, in my opinion, be improved: presenting all the tasks first and then all the results delays **the key findings until relatively late in the paper** (around page 5 and beyond).

**Significance.** The paper tackles a timely question, grokking, which is currently central to discussions in many areas of machine learning. In particular, grokking challenges classic theories of learning that suggest test performance should eventually decrease as a model overfits its training data, making it an important new direction of research. By generalizing the tasks commonly used in this research area, the authors position their work well within the existing literature, and **the paper would generally be of interest to the ICML audience.**

**Originality.** Several papers have already looked at how symmetry affects generalization. The main novelty of this work is that they go beyond modular algebra tasks classically used. This is, in my opinion, a sufficient novelty in itself. However, the breadth of analyses performed on these new tasks is relatively limited. This is why I do not provide a higher originality score; **the authors mainly use common tools, but applied to new tasks.**

---

> ### Author Rebuttal · Authors · 2026-03-31
>
> We sincerely thank the reviewer for the insightful comments.
>
> Below we have tried our best to address your concerns and questions.
>
> For Figures, we ask you to refer to this link. Link: https://anonymous.4open.science/r/ICML_2026_supp-D97F
>
> > **1. Clarification of theoretical contribution**
>
> We thank the reviewer for this important comment. We agree that the statements in Propositions 7.1–7.3 are elementary and closely related to classical results in topology, distance geometry, and graph realization.
> For Proposition 7.1 (modular arithmetic), we cited references in the appendix, as it follows from well-known results in representation theory. For clarity, we will move these references and a brief explanation to the main text.
>
> For Propositions 7.2–7.3, to the best of our knowledge, there is no single reference that states these results in the exact form used here, particularly under the combination of partial constraints and symmetry assumptions. Proposition 7.2 is closely related to classical results in distance geometry (e.g., [1]), where realizations from edge-length constraints are well understood and unique up to rigid transformations. Proposition 7.3 is related to classical representation results in order and measurement theory (e.g. [2]), where transitive relations admit real-valued representations. However, these works do not explicitly address the embedding formulations considered here. Our intention is not to claim novelty of these individual facts, but to provide a self-contained and unified formulation tailored to the embedding perspective studied in this work.
>
> [1] Crippen & Havel. Distance geometry and molecular conformation.
>
> [2] Krantz, et al.  Foundations of measurement.
>
> ---
>
> > **2. Discussion of possible classes of symmetries.**
>
> We thank the reviewer for this insightful question. A general approach is to characterize symmetries as a transformation group $G$ acting on the input space $\mathcal{X}$, where the target function satisfies an invariance condition:
> $f(Tx) = f(x) \quad \text{(invariance)},$
> for $T \in G$. From this perspective, one can systematically search for symmetries by identifying transformations $T$ that preserve the model outputs. While it is not possible to capture all symmetry, it can serve as a starting point for searching for symmetries.
>
> For example, we additionally experiment with counting task $[c1, c2, c3, q]$ where the output is to count the number of $q$ appearing in context.  Applying positional permutation shows clear division in symmetric KL, and allows us to find S3 positional invariance, as shown in `Reviewer_rWvz/counting.png`
>
> ---
>
> > **3. Link to the study of optimization dynamics**
>
> Thank you for the insightful comment. We agree that both optimization and representation perspectives are important and complementary: the former explains training dynamics, while the latter explains what structures enable generalization.
> Prior works such as [1,2] provide elegant theoretical analyses of optimization dynamics in simplified settings. However, these analyses are often difficult to extend to more complex architectures and tasks. Our work instead focuses on understanding generalization in such settings, where precise theoretical analysis remains challenging, and thus adopts an empirical approach.
>
> To connect to training dynamics, we analyze gradients during training modular addition and find that during the symmetry acquisition phase, the gradient induced by symmetry constraints aligns with the optimizer gradient (positive cosine similarity). This suggests that the model implicitly learns symmetry structures that drive generalization.
> We will clarify these connections and discuss them more explicitly in the revision. ( For the visualization of the gradients, please refer to `Reviewer_rWvz/grad.png`)
>
> ---
>
> > **4. Link to manifolds and Lie groups**
>
> We thank the reviewer for this insightful question. We distinguish two types of symmetries in our framework. For algebraic tasks such as modular arithmetic, the intrinsic symmetries (commutativity, associativity) define an abelian group, which admits a continuous realization (e.g., $ \mathbb{Z}_p \subset S^1$) and corresponds to geometric symmetries such as translations studied in Riemannian settings. However, many symmetries we consider (e.g., transitivity in comparison tasks or triangle equality in graph metric completion) do not define group actions and therefore do not correspond to isometries of a manifold. Instead, they impose relational consistency constraints that still organize the embedding space geometrically, but without inducing global invariances. Our framework thus extends beyond the symmetry notions typically considered in Riemannian geometry.
>
> ---
>
> > **5. Negative Penalty**
>
> We thank the reviewer for this valuable suggestion. We include the run result below, where negative penalty prevents generalization. (Please refer to the images inside `Reviewer_rWvz/neg_penalty` directory)

---

> > ### Author Rebuttal · Reviewer_rWvz · 2026-04-03
> >
> > I thank the authors for their response. My concerns have been addressed.
> >
> > The theoretical contributions, which occupy a substantial portion of the paper, appear limited, as the authors themselves acknowledge that most of the results are already established.
> >
> > Because of this will keep my current score.

---

> > > ### Author Response · Authors · 2026-04-08
> > >
> > > We thank the reviewer for the thoughtful evaluation and for acknowledging that the concerns have been addressed. We also appreciate the positive assessment of the paper’s significance and presentation.
> > >
> > > We would like to clarify the intended scope of our contribution. The primary goal of this work is **empirical and conceptual**, rather than formal theoretical development. Specifically, we identify intrinsic task symmetry as a unifying driver of generalization, and support this claim through systematic experiments across multiple domains (algebraic, structural, and relational), together with a proposed three-stage training dynamics of grokking. We believe this type of contribution aligns with prior impactful works that introduce new perspectives and phenomena, even when a full theoretical characterization is beyond the scope of the work.
> > >
> > > Regarding theory, we agree that individual components of Propositions 7.1–7.3 are closely related to classical results. However, our contribution lies in **establishing a concrete connection between intrinsic symmetries and emergent embedding geometries**, and in showing how this connection explains generalization behavior across tasks. To our knowledge, this symmetry → geometry → generalization perspective has not been previously demonstrated in a unified manner.
> > > In addition, the rebuttal provides further evidence addressing mechanism and causality, including:
> > >
> > >  (i) gradient alignment analysis showing that symmetry directions are actively learned during training,
> > >
> > >  (ii) negative symmetry experiments demonstrating that suppressing symmetry impairs generalization, and
> > >
> > >  (iii) a new theoretical result showing that symmetry combined with partial observations suffices for full generalization in cyclic groups (in response to Reviewer pYGF).
> > >
> > > Given that the reviewer acknowledges the significance of the work and that the main remaining concern relates to theoretical depth, we hope the contribution can be evaluated in light of its intended empirical and conceptual scope.
> > > We thank the reviewer again for the constructive feedback.

---

### Official Review · Reviewer_YrPA · 2026-03-12

**Soundness:** 3
**Presentation:** 3
**Significance:** 2
**Originality:** 2
**Overall Recommendation:** 4
**Confidence:** 3

**Summary:**

This paper analyzes the emergence of "grokking" in neural networks, proposing a three-stage process: 1) Memorization, 2) Symmetry Acquisition, and 3) Convergence to a low-rank geometric representation. By analyzing the curves of standard objectives and symmetry acquisition loss, the authors demonstrate that test accuracy increases when the model begins to learn common conceptual symmetries. Furthermore, they show that generalization stabilizes as the geometric representation converges to a low-rank state. The paper also provides theoretical insights into the geometric forms underlying different arithmetic tasks and suggests methods to accelerate the grokking phenomenon.

**Compliance With Llm Reviewing Policy:**

Affirmed.

**Final Justification:**

The concerns raised in my “Rebuttal Acknowledgement by Reviewer YrPA” have not been sufficiently resolved. Therefore, I am maintaining my original score.

**Key Questions For Authors:**

- Please address the points raised in the Weaknesses section.

**Limitations:**

The study lacks validation on large-scale models like Transformers and does not address real-world scenarios or complex arithmetic tasks where symmetry information is unknown or mixed.

**Strengths And Weaknesses:**

### Strengths

- **(Originality & Significance)** Unlike previous studies, this work experimentally demonstrates that grokking initiates through the learning of common concepts during the symmetry acquisition phase. The authors show that providing explicit objectives for symmetry can accelerate grokking.

- **(Soundness)** The paper provides a theoretical basis for geometric observations in arithmetic tasks and shows that representations become isomorphic during the geometric organization phase.

---

### Weakness (Major Issues)
1. **Soundness & Generalizability**
    - **Practicality of Symmetry Objectives**: If the symmetry information of an arithmetic task is unknown a priori, it seems impossible to provide the auxiliary objectives suggested. The paper lacks a discussion on scalable or unsupervised methods for identifying these symmetries.
    - **Dataset-Specific Inductive Bias**: The verification appears to rely on datasets where a strong inductive bias for symmetry is already embedded in the construction. It raises the question: if a task involves multiple overlapping symmetries, how would the geometric representation emerge and organize?
    - **Inconsistency in Results**: There appears to be a discrepancy between the Lattice results in Figure 7 and the findings presented in Appendix B. Please clarify this inconsistency.
    - **Model Generalization (Transformer)**: In Section 8.4, the authors state that grokking occurs in MLPs as well as Transformers. However, given that Transformers are the primary architecture where grokking is studied, it is essential to verify if the observed phenomena (and the acceleration of grokking via symmetry objectives) hold true for Transformers. This would demonstrate the applicability of the findings to more general models and tasks.

2. **Presentation**
    - **Reproducibility**: In Section 8.1, the specific implementation details of $L_{aux}$ for each task are missing. This makes it difficult to reproduce the experiments and verify the claimed acceleration.

---

---

> ### Author Rebuttal · Authors · 2026-03-31
>
> We sincerely thank the reviewer for the insightful comments.
>
> Below we have tried our best to address your concerns and questions.
>
> For Figures, we ask you to refer to this link. Link: https://anonymous.4open.science/r/ICML_2026_supp-D97F
>
>
> ---
>
> > **W1. Practicality of Symmetry Objectives**
>
> Thank you for the valuable comment. We agree that requiring explicit knowledge of task symmetries may limit the direct applicability of symmetry-based auxiliary objectives unless known a priori.
>
> Our primar goal in this work is to establish a causal role of intrinsic atsks symmetry in driving generlaization during grokking. To this end, we intentionally study settings where the symmetry structure is known, and show that enforcing it as a regularizer can accelerate generaliaation. We wil lclarify this scope more explicitly in the revision.
>
> That said, we do not view manually specified symmetry objectives as the only possible route. In cases where the exact symmetry is unknown, a more scalable alternative may be to encourage the emergence of structured embedding geometry, which in our experiments is closely associated with generalization.
>
> ---
>
> > **W2. Dataset-Specific Inductive Bias**
>
> Thank you for the insightful comment. Our goal is to understand how models learn and exploit intrinsic symmetries present in the dataset. Importantly, the tasks we consider already involve multiple overlapping symmetries. For example, modular arithmetic includes commutativity and associativity, graph completion involves triangle equality together with symmetry, and comparison tasks combine transitivity with anti-commutativity although we do not present second symmetry in graph/comparison tasks in the paper. We believe that there could be another symmetry within tasks.
>
> In our experiments, we observe that these symmetries are jointly acquired during training. We hypothesize that these symmetries are mutually consistent and collectively define the underlying task structure - and generalization emerges when the model aligns with this combined symmetry structure.
>
> Regarding tasks with multiple overlapping symmetries, our results suggest that the learned geometric representation organizes in a way that simultaneously respects all compatible symmetries. This is consistent with our observation that symmetry acquisition precedes geometric organization. On the other hand, if a task contains fundamentally incompatible or contradictory symmetries, we expect that generalization to likely fail.
>
> ---
>
> > **W3. Inconsistency in Results**
>
> Thank you for raising this point. We want to first clarify that the **results are consistent**, yet the apparent discrepancy arises because Figure 7 and Appendix B use **different x-axis scalings**, which can make the trends look misaligned at first glance. When comparing corresponding epochs, one would then entice that results do agree. To avoid this confusion, we will revise the figures to use consistent scaling.
>
> ---
>
> > **W4. Model Generalization**
>
> Thank you for raising this point. We would first like to respectfully clarify that the main results in Section 8.4, including Figure 7, are already for the **Transformer** setting. Thus, the phenomenon of grokking, as well as its acceleration through symmetry objectives, is indeed verified in Transformers in our paper.
>
> The purpose of including Figure 8 is different: it is intended to show that this acceleration can also occur in MLP and therefore is invariant across model architectures. In other words, while Section 8.4 already establishes the effect in the primary architecture of interest, Figure 8 further demonstrates that the same phenomenon also extends beyond Transformers.
>
> If the reviewer is specifically asking whether grokking without weight decay, together with symmetry-based acceleration, can occur in Transformers, we also included an additional Transformer experiment below.
>
> Figure: `Reviewer_YrPA/transformer_wo_wd_runs.png` in the link
>
>  In this experiment, we follow the setting of [1], which showed that grokking is possible in Transformers without weight decay when floating-point errors in the Softmax function are mitigated. Under this setting, we find that adding the symmetry-prompting regularizer again significantly accelerates generalization.  We will include this figure in the final version to make this point clearer.
>
> ---
>
> > **W5. Reproducibility**
>
> Due to space limitations, we respectfully refer the reviewer to Appendix A.1, A.2, and A.3 for the detailed formulations of the **symmetry-prompting** $L_{aux}$ across each domain. For **geometry-prompting** $L_{aux}$, we directly regularize the embedding space:
>
> $$
> \mathcal L_{\mathrm{nuc}}=|E|*,\quad
> \mathcal L_{\mathrm{lip}}=\tfrac1V\sum_i |E_i-E_{i+1}|_2^2,\quad
> \mathcal L{\mathrm{ent}}=-\sum_i \pi_i \log \pi_i,
> $$
>
> We will include these details in the appendix in the final version and make the connection explicit in the main text.
>
> ---
>
> [1] GROKKING AT THE EDGE OF NUMERICAL STABILITY (Prieto, 2025)

---

> > ### Author Rebuttal · Reviewer_YrPA · 2026-04-03
> >
> > I still have some concerns regarding the analysis of the Transformer architecture. In particular, the first issue **is quite critical**. In the newly provided experimental results, the modular addition test accuracy appears to reach 100% well before 100 epochs. In light of this, I find it very difficult to reconcile these results with Figure 7 (modular addition, Transformer), which still shows unsolved cases even at 10k (10,000 epochs). This discrepancy is highly unusual and, in my view, requires a much more detailed explanation. At a minimum, the paper should provide the exact experimental setup and a clearer discussion of how these seemingly inconsistent results arise.
> >
> > In addition, I am not fully convinced by the claim of invariance across model architectures based on the MLP results in Figure 8. Even a simple two-layer MLP can already achieve close to 100% performance on this task, so it remains unclear to me how these results substantiate invariance across architectures. Moreover, a Transformer is still composed of linear layers and attention blocks, and therefore does not depart sufficiently from the MLP setting to strongly support such a conclusion. To make this claim more convincing, I believe that validation on at least one substantially different architecture, such as a CNN-based model, would be necessary.

---

> > > ### Author Response · Authors · 2026-04-08
> > >
> > > We sincerely thank the reviewer for the insightful comment.
> > >
> > > **A) Regarding the Discrepancy**
> > >
> > > Upon re-checking the rebuttal experiment, we found that the run had inadvertently been executed with a MLP rather than the intended Transformer. We sincerely apologize for this error.
> > >
> > > We therefore re-ran the experiment using the correct Transformer architecture and report the updated results below (mean ± std over 5 runs, 30k epochs). Here, **(p=0.3)** is the training-fraction setting used throughout the main paper, and we additionally evaluate **(p=0.8)** because, in the absence of weight decay, smaller training fractions took considerably longer to achieve perfect generalization.
> > >
> > > We further note that training without weight decay introduces numerical instability in Softmax in transformer architecture as mentioned in [1], and thus we use StableMax instead of Softmax. Please note that below MLP setting differs from the setting in Figure 8 which does not utilize StableMax.
> > >
> > > ---
> > >
> > > **From Rebuttal (MLP with StableMax)**
> > >   | Weight Decay ($w$) | Regime | p=0.8 |
> > >   |---|---|---|
> > >   | $0.0$ | baseline | $5764.4 \pm 1156.4$ |
> > >   |  | **+ symmetry** | $54.0 \pm 2.9$ |
> > >
> > > ---
> > > **Corrected result (Transformer):**
> > >
> > > | Weight Decay ($w$) | Regime     |                $p=0.3$ |              $p=0.8$ |
> > > | ------------------ | ---------- | ---------------------: | -------------------: |
> > > | $1.0$              | baseline   |     $7991.0 \pm 814.4$ |   $1515.2 \pm 231.9$ |
> > > |             | + **symmetry** |  **$1328.4 \pm 73.5$** | **$397.0 \pm 15.5$** |
> > > | $0.0$              | baseline   |            Not reached |     $318.2 \pm 35.9$ |
> > > |              | + **symmetry** | **$1213.6 \pm 102.1$** | **$224.0 \pm 14.2$** |
> > >
> > > ---
> > > Across all settings, adding the symmetry regularizer (commutativity + associativity) reduces the steps required for perfect generalization.
> > > **Overall, the earlier difference between the unsolved 10k case and the shorter runs is explained by differences in training fraction and by whether the symmetry regularizer is applied.**
> > >
> > > We provide the per-seed raw epoch counts and the corresponding plots for these results in the below link: https://anonymous.4open.science/r/ICML_2026_supp_pt2-AD33.
> > >
> > > We will revise the manuscript to make the hyperparameter settings explicit and avoid further confusion. We thank the reviewer again for the helpful comment and hope this clarification resolves the concern.
> > >
> > > **B) Regarding different models**
> > >
> > > We thank the reviewer for the helpful suggestion. Our main intention was to show that the observed effect is not specific to Transformers. Since our setting focuses on sequence modeling, Transformers are the primary architecture of interest, and we included the MLP result as an additional non-attention baseline.
> > >
> > > At the same time, as the reviewer points out, we did not evaluate all model classes, so we will revise the wording to make the scope of the claim more precise. We appreciate the reviewer’s comment and will consider testing the generality of the phenomenon on more diverse architectures in the future work.
> > >
> > >
> > > ---
> > >
> > > [1] GROKKING AT THE EDGE OF NUMERICAL STABILITY (2025)

---

### Official Review · Reviewer_pYGF · 2026-03-13

**Soundness:** 3
**Presentation:** 4
**Significance:** 3
**Originality:** 3
**Overall Recommendation:** 5
**Confidence:** 3

**Summary:**

This paper proposes that grokking, the sudden transition from memorization to generalization in neural networks, results from "intrinsic task symmetries". The authors analyze three classes of tasks (modular arithmetic, graph metrics, relational tasks) and report a consistent three-stage training dynamic: initial memorization of the data, acquisition of intrinsic symmetries in the model’s representation, and geometric organization of the embedding space into a low-dimensional structure. They show that generalization coincides with the second stage and leverage these findings to design a symmetry-based generalization criterion and two auxiliary-loss strategies ("symmetry-prompting" and "geometry-prompting") that accelerate generalization.

**Compliance With Llm Reviewing Policy:**

Affirmed.

**Final Justification:**

The paper addresses all of my questions and provides a better context for the problem statement.

**Key Questions For Authors:**

1. How is Figure 6 and 8 generated? My understanding is that grokking can have high variability between runs. Is the experiment repeated over multiple runs? There should be error bars if so.
2. Can you provide some quantitative measures of the results? It should demonstrate statistical significance to support the hypothesis.
3. What are the formulas for symmetry-prompting and geometry-prompting losses?

**Limitations:**

The paper should clarify the scope within which they can make the claim that "intrinsic symmetry drives generalization".

**Strengths And Weaknesses:**

Strength:

The idea of a causal link between latent task symmetries and grokking is novel. While prior work has observed grokking in the relevant tasks (such as modular arithmetic), this is the first paper that justifies the causal relation between them.

For the intrinsic task symmetries it defines and the corresponding synthetic dataset, the paper provides comprehensive experimental evaluations as evidence of the causal connection. The resulting prior prompting methods are effective in accelerating generalization.

Weakness:

The paper lacks formalism that enables the generalization of intrinsic task symmetry beyond the listed task. And as the papers are largely empirical, it lacks a formal theoretical analysis proving that symmetry acquisition necessarily precedes or causes generalization. Given that the empirical results are over a synthetic dataset with relatively simple data with simple structures, it's also unclear how much this causal effect translates to grokking in general datasets.

Some experimental details are missing, and the results are largely qualitative (or are not quantified transparently). (See questions)

---

> ### Author Rebuttal · Authors · 2026-03-31
>
> We sincerely thank the reviewer for the insightful comments.
> Below we have tried our best to address your concerns and questions.
>
> For Figures, we ask you refer to the below link.
> Link: https://anonymous.4open.science/r/ICML_2026_supp-D97F
>
> ---
>
> > **W1: Lack of Formalism and Theoretical Analysis**
>
> We thank the reviewer for raising this important point. To partially address this concern, we note that for structured tasks such as modular addition/multiplication, one can explicitly establish a theoretical link: symmetry constraints (e.g., commutativity and associativity), together with partial memorization, can be sufficient to reconstruct the full operation. This provides a concrete instance where symmetry enables generalization from a small subset of observed pairs.
>
> In particular, under the assumption that the operation forms a cyclic group, one can show that only a vanishing fraction of the Cayley table is needed to recover the entire operation:
>
> **Statement** Let $G=\{0,1,\dots,p-1\}$ and let $\odot$ be a binary operation on $G$.
> Assume that $(G,\odot)$ is a cyclic group of order $p$, with identity element $0$ and generator $1$.
> Then there exists a subset
> $M \subset G \times G,$ with $|M| = 2p-1,$ such that the values of $\odot$ on $M$, namely
> $\tilde{M} :=  \\{ (a,b,c) \in G^3 : (a,b)\in M,  a \odot b = c  \\} ,  $
> uniquely determine the full Cayley table of $(G,\odot)$. Consequently, the fraction of entries required to reconstruct the operation is
> $$
> \frac{|M|}{|G\times G|} = \frac{2p-1}{p^2} \sim \frac{2}{p}.
> $$
>
> > **W2: Empirical results over synthetic datasets & Uncertainty on how it translates to grokking in general datasets:**
>
> We agree that extending this analysis to more complex real-world datasets is an important direction for future work. At the same time, we use controlled synthetic settings for grokking because they allow the mechanism of generalization to be studied cleanly. We respectfully suggest our structural and relational tasks partially capture aspects of real-world structure, including compositions and knowledge-graph-like reasoning, therefore position these results as a controlled but meaningful step toward understanding generalization in more realistic domains.
>
> ---
>
> > **Q1. How is Figure 6 and 8 generated? My understanding is that grokking can have high variability between runs ...**
>
> Thank you for raising this point. We agree that grokking can exhibit substantial run-to-run variability and that this should be reflected more clearly. In the current submission, Figures 6 and 8 show only single-run results. To address this, we have additionally attached visualization for multi-run results in the link below.
>
> **Figure 6**
> - Please refer to the run results inside `Reviewer_pYGF/{comparison, lattice, task1}` in the provided link.
>
> **Figure 8**
> - Please refer to the run result inside `Reviewer_pYGF/nowd_modular/run1.png`
>
> For Figure 8, we have revised the figure to report the mean across 5 runs together with error bars, and we will update the main paper accordingly.
>
> For Figure 6, while we also generated the corresponding multi-run version, directly adding error bars to the main figure substantially reduces readability because of the variability in the loss and symmetry metrics. We therefore plan to keep the current single-run visualization in the main text for clarity, and include the multi-run version in the appendix so that variability is also properly documented.
>
> We thank the reviewer again for this helpful suggestion.
>
> ---
>
> > **Q2. Can you provide some quantitative measures of the results? It should demonstrate statistical significance to support the hypothesis.**
>
> Below we provide a quantitative analysis corresponding to Figure 5, aimed at testing the soft-threshold behavior of the symmetry metric. For each run, we compare the symmetry metric at the first checkpoint with perfect generalization against the minimum symmetry metric among checkpoints with acc $\le$ 0.99. We then test the null hypothesis that the latter is no larger than the former.
>
> Across 20 runs, we obtain:
>
>  * **Modular Addition:** $p = 9.5367 \times 10^{-7}$
>  * **Lattice:** $p = 0.02734$
>  * **Comparison (2D):** $p = 0.03125$
>
> ---
>
> > **Q3. What are the formulas for symmetry-prompting and geometry-prompting losses?**
>
> Due to space limitations, we respectfully refer the reviewer to **Appendix A.1, A.2, and A.3** for the full formulations of the symmetry-prompting losses across tasks.
>
> For geometry-prompting losses, we directly regularize the embedding space (shared across domains), e.g.
>
> $$
> \mathcal L_{\mathrm{nuc}}=|E|*,\quad
> \mathcal L_{\mathrm{lip}}=\tfrac1V\sum_i |E_i-E_{i+1}|_2^2,\quad
> \mathcal L{\mathrm{ent}}=-\sum_i \pi_i \log \pi_i,
> $$
>
> encouraging low-rank, smooth, and well-distributed representations.
> We will include these details in the appendix and make the reference to it explicit in the main text.
>
> ---
>
> We thank the reviewer again for the valuable comments.

---

> > ### Author Rebuttal · Reviewer_pYGF · 2026-04-06
> >
> > I thank the authors for their response. The authors have addressed all of my questions. I have raised my score.

---

> > > ### Author Response · Authors · 2026-04-08
> > >
> > > We sincerely thank the reviewer for the positive re-evaluation and for confirming that our responses addressed all concerns.

---

### Official Review · Reviewer_zA3s · 2026-03-13

**Soundness:** 3
**Presentation:** 3
**Significance:** 3
**Originality:** 2
**Overall Recommendation:** 4
**Confidence:** 1

**Summary:**

Through experiments and theoretical analysis on Modular Arithmetic, Graph Completion, and Comparison tasks, this work demonstrates that during training, neural network embeddings develop symmetry and collapse into lower-dimensional structures. Furthermore, the authors propose a loss function designed to encourage such symmetry and accelerate optimization. These findings suggest that the intrinsic symmetry of a task is a key driver of grokking.

**Compliance With Llm Reviewing Policy:**

Affirmed.

**Final Justification:**

All my concerns are solved. I raise my score from 3 to 4.

**Key Questions For Authors:**

1. How will general language modeling task benefit from this work?
2. Will this phenomenon also appears in transformer rather than MLP?

**Limitations:**

yes

**Strengths And Weaknesses:**

Strength:
1. Soundness. The comprehensive experiments including embedding visualization, rank measure, and task design clearly verify that the acquiring symmetry is vital for generalization.
2. Presentation. Task settings and background are clearly illustrated in section 3.

Weakness:
1. The proposed tasks are very simple toys and far from the interesting grokking of LLM.
2. Symmetry helps generalization is a well-known conclusion [1]. How this work goes beyond this conclusion?

[1] Bryn Elesedy. Group Symmetry in PAC Learning.

---

> ### Author Rebuttal · Authors · 2026-03-31
>
> Dear Reviewer, we sincerely thank you for your careful review and comments.
>
> Below, we have addressed your concerns and questions.
>
> ---
>
> > **W1. The proposed tasks are very simple toys and far from the interesting grokking of LLM.**
>
> We thank the reviewer for the question. We would like to clarify that the use of “simple” algorithmic tasks is a *standard choice* in the grokking literature, as such controlled settings are essential for isolating mechanisms of generalization. Prior work has primarily studied grokking in relatively simple domains (e.g., modular arithmetic), where careful analysis is possible.
>
> Our work extends this line of research by considering multiple domains (algebraic, structural, and relational) and showing that a single mechanism, intrinsic task symmetry consistently explains generalization across them . In this sense, our contribution lies not in task complexity, but in identifying a *unified principle* that holds beyond a single setting.
>
> Regarding relevance to language models, our focus is on understanding how models acquire algorithmic structure, which is closely related to known limitations of LLMs (e.g., numerical reasoning errors and hallucination). Our results suggest that generalization depends on learning underlying symmetries rather than memorization, offering insight into these failure modes.
> Furthermore, we translate this understanding into practice by proposing symmetry-based training strategies that accelerate generalization , which we believe provides a principled direction for improving algorithmic reasoning in language models.
>
> ---
>
> > **Q1. Symmetry helps generalization is a well-known conclusion [1]. How this work goes beyond this conclusion?**
>
>
> We thank the reviewer for this point. We agree that prior work, including [1], establishes that **symmetry can improve generalization when it is built into the model (e.g., invariant/equivariant hypotheses)** by reducing effective sample complexity .
>
> Our work differs in a fundamental way: rather than assuming symmetry as prior knowledge, we study **how symmetry is acquired during training** and how this process drives generalization. In particular, we show that symmetry emerges as a distinct phase (symmetry acquisition) that coincides with grokking and leads to structured representations.
>
> Thus, while prior work explains the benefit of pre-specified symmetry, our contribution is to explain **the mechanism by which symmetry is learned and how it dynamically induces generalization**, as well as how to leverage this insight for accelerating training.
>
> ---
>
> > **W2, Q2. Will this phenomenon also appears in transformer rather than MLP?**
>
> We would like to clarify that grokking is generally understood as a phenomenon that is not tied to a specific architecture.
>
> In our experiments, we primarily adopt a 1-layer transformer architecture, following the standard setup used in prior grokking studies to ensure consistency with existing literature. To further check whether our observations depend on architectural choice, we additionally include an MLP experiment (Figure 8). We observe similar behavior in this setting as well, suggesting that the phenomenon we study is not specific to transformers.
>
> We will revise the manuscript to make this clarification more explicit and avoid potential misunderstanding.
>
>
> [1] Bryn Elesedy. Group Symmetry in PAC Learning.

---

> > ### Author Rebuttal · Reviewer_zA3s · 2026-04-01
> >
> > The detailed feedback solves all my concerns. I will raise my score to 4.

---

> > > ### Author Response · Authors · 2026-04-08
> > >
> > > We sincerely thank the reviewer for the re-evaluation of our submission and for the positive assessment.

---

### Decision · Program_Chairs · 2026-04-30

**Decision:**

Accept (regular)

**Comment:**

The authors propose that intrinsic task symmetries drive grokking via a three-stage training dynamic (memorization → symmetry acquisition → geometric organization), supported by experiments on modular arithmetic, graph metric completion, and comparison tasks, and extended by two auxiliary-loss interventions - symmetry-prompting and geometry-prompting - that measurably accelerate generalization. Three reviewers endorse the framing as timely and well validated; the symmetry-based generalization criterion and prompting losses give the analysis actionable teeth. The dissent (rWvz, confidence 4, Weak Reject maintained post-rebuttal) is principled: the theoretical section is substantial but phenomenological rather than mechanistic, and the causal claim would be stronger with a symmetry-suppression experiment rather than only symmetry-promotion. Other concerns include toy / algorithmic scope (no transformers or LM evaluation), missing error bars, inconsistency between Figure 7 and Appendix B, and sharper positioning versus Elesedy-style group-symmetry literature. The rebuttal raised zA3s's score (3 → 4); rWvz's theoretical concerns persisted; pYGF and YrPA maintained positive scores. The raw mean of 4.0 overstates support - the 5 sits at confidence 3, the 3 at confidence 4 - but the empirical package is strong and the framing novel enough that I land at poster-level Accept. For camera-ready, the authors should add error bars, explicit per-task auxiliary-loss formulas, the Figure 7 / Appendix B fix, either a symmetry-suppression experiment or a frank limitations paragraph, sharper positioning versus prior work, and a scalability discussion on estimating symmetries when ground truth is unknown.